# Basal-to-inflammatory transition and tumor resistance via crosstalk with a pro-inflammatory stromal niche

Nancy Yanzhe Li [1], Weiruo Zhang [2], Daniel Haensel [1], Anna R. Jussila [1], Cory Pan [1], Sadhana Gaddam[1], Sylvia K. Plevritis[2] & Anthony E. Oro [1] ✉

Cancer-associated inflammation is a double-edged sword possessing both pro- and anti-tumor properties through ill-defined tumor-immune dynamics. While we previously identified a carcinoma tumor-intrinsic resistance pathway, basal-to-squamous cell carcinoma transition, here, employing a multipronged single-cell and spatial-omics approach, we identify an inflammation and therapy-enriched tumor state we term basal-to-inflammatory transition. Basal-to-inflammatory transition signature correlates with poor overall patient survival in many epithelial tumors. Basal-to-squamous cell carcinoma transition and basal-to-inflammatory transition occur in adjacent but distinct regions of a single tumor: basal-to-squamous cell carcinoma transition arises within the core tumor nodule, while basal-to-inflammatory transition emerges from a specialized inflammatory environment defined by a tumor-associated TREM1 myeloid signature. TREM1 myeloid-derived cytokines IL1 and OSM induce basal-to-inflammatory transition in vitro and in vivo through NF-κB, lowering sensitivity of patient basal cell carcinoma explant tumors to Smoothened inhibitor treatment. This work deepens our knowledge of the heterogeneous local tumor microenvironment and nominates basal-to-inflammatory transition as a drug-resistant but targetable tumor state driven by a specialized inflammatory microenvironment.

While advances in therapeutic strategies including chemotherapy, targeted therapy, radiotherapy, and immunotherapy have equipped clinicians with a diverse arsenal to combat cancer, the efficacy of these treatments remains inconsistent and ineffectual[1]. Initial successes in naïve tumor size reduction are thwarted by resistant tumor outgrowth, highlighting the need to understand tumor heterogeneity. In addition to the intrinsic mechanisms underlying cancer cell evasion, the complex tumor microenvironment (TME) orchestrated by tumor cells has been increasingly shown to restrict the effectiveness of cancer treatments and impact patient survival[2–4]. Single-cell genomic technologies have revealed that resistance stems from extensive tumor and tumor-associated immune heterogeneity and plasticity, with the dynamics between microenvironment and drug interventions and tumor evolution programs facilitating therapy circumvention[5–8].

A particular gap in knowledge surrounds the local interactions between tumors and inflammation in the same tumor subregion. Tumor-associated inflammation remains a double-edged sword, possessing both pro- and anti-tumor properties through ill-defined tumor-immune dynamics that belie the common classification of hot and cold immunoreactive tumors[9–12]. A growing body of literature implicates the inflammatory response in tumor protection, progression and drug resistance where tumors surrounded by inflammation correlate with poor patient outcomes[9,13,14]. Drug responses change dramatically within these milieus, thwarting effective responses to portions of the

[1]Program in Epithelial Biology, Stanford University School of Medicine, Stanford, CA, USA. [2]Department of Biomedical Data Science, Stanford University School of Medicine, Stanford, CA, USA. ✉e-mail: oro@stanford.edu

tumor[15]. Differences in immune landscapes between human and murine experimental systems further added a layer of complexity to investigations centered around immune interplay with drug responses, often resulting in paradoxical or false negative findings. Therefore, there is a need for a higher resolution of both the human patient tumor epithelial response and the profiles of tumor-adjacent inflammatory cell types, as well as the need to dissect the signaling interactions between the two.

Basal cell carcinomas (BCC) are the ideal experimental system to study tumor-immune dynamics and resistance due to relative simplicity, high incidence, and ability to re-biopsy patients. BCCs arise through tumor epithelial hedgehog pathway upregulation within wound-like inflammation, with the tumor instructing the surrounding inflammatory environment to induce TREM2 + VCAM1 + myeloid cells that lack mature macrophage or dendritic markers[16]. BCCs and TREM2 + VCAM1+ myeloid cells subsequently establish a self-propagating tumor-immune niche where both BCCs and myeloid cells proliferate stoichiometrically to maintain Smoothened inhibitor (SMO[i])-sensitive tumor growth and retain the ability to recruit and instruct additional bone marrow-derived monocytes as BCCs enlarge.

As BCC sensitivity to targeted SMO[i] therapy remains variable, we have focused our investigations on BCC tumor plasticity pathways that are selected for and arise during drug treatment. We previously identified reversible basal-to-squamous cell carcinoma transition (BST) as a tumor cell-intrinsic mechanism utilized by tumor cells to achieve resistance to SMO[i] vismodegib[17,18]. Squamous changes occur within tumor nodules away from the tumor microenvironment and are driven by inducible expression of combinatorial AP-1 family member expression in vitro and in vivo, with c-FOS driving BCC's phenotypic transitions away from basal epithelial state toward well-differentiated squamous cell carcinomas (SCC). BST appears spontaneously in naïve sporadic BCCs and becomes enriched in drug-treated resistant tumor populations. While the molecular basis of this BCC tumor-intrinsic pathway has been elucidated, tumor resistance states driven through the complex dynamics of tumor-extrinsic pathways via interactions with the inflammatory response remain poorly studied, motivating further interrogation.

Here, employing a multipronged single-cell sequencing, spatial transcriptomics, and multiplexed imaging approach along with functional validation, we identify a surprising inflammation and therapy-enriched tumor epithelial cell state we term basal-to-inflammatory transition (BIT). BIT signature correlates with poor overall patient survival in many other epithelial tumors. BST and BIT occur in adjacent but distinct regions of a single tumor: BST arises within the core tumor nodule, while BIT emerges from a specialized inflammatory environment defined by a tumor-associated TREM1 myeloid signature. We show that TREM1 myeloid-derived cytokines IL1 and OSM activate the inflammatory NF-κB family of transcription factors within the BIT tumor epithelium and lower the sensitivity of human BCC explant tumors to SMO[i] treatment, conferring resistance. This work deepens our knowledge of the heterogeneous local tumor microenvironment and nominates BIT as a drug-resistant but targetable tumor state driven by a specialized inflammatory microenvironment.

## Results

### Integrated single cell analysis of human BCC tumors identifies a tumor state (TS3) enriched by drug treatment

To provide a high-resolution definition of human BCC patient tumor samples at the molecular and spatial level, we employed a multi-modal approach combining human tumor single-cell RNA sequencing (scRNA-Seq), single-cell chromatin accessibility (scATAC-Seq) study, multiplexed imaging, and spatial transcriptomics for integrative analysis of both single-cell dissociated tumor samples and matched formalin-fixed paraffin-embedded (FFPE) tumor tissues. We subsequently employed a cancer cell line, mouse tumor model, and patient-derived organoids to validate our findings (Fig. 1a, Supplementary Fig. 1a). To better understand the dynamic states of BCC tumor epithelium pre- and post-SMO[i] therapy, we integrated the scRNA-Seq data of tumor epithelial populations from two treatment-naïve patients with that of two SMO[i]-treated patients[19] from Yost et al. 2019, (Supplementary Figs. 1b, c) and identified 6 distinct tumor states (Fig. 1b). Notably, among BCC tumor states, tumor state 3 was enriched in SMO[i]-treated BCC tumors relative to naïve BCC tumors (Fig. 1c). Tumor state 3 (TS3) is defined by cluster markers CHI3L1, TAGLN, ITGAV and VCAM1, along with a decrease in expression level of GLI1 (Fig. 1d), a canonical driver of hedgehog signaling in BCC tumors, typically associated with a switch away from dependency on this signaling pathway.

To understand whether TS3 typically pre-exists in human BCCs prior to drug treatment, we examined subpopulations of treatment naïve tumor epithelium as sources of SMO[i]-enriched tumor state (Fig. 1e). As expected, we observed a tumor cluster with markers (TACSTD2, LYPD3, LY6D, CLDN4, KRT6A, KRTDAP) characteristic of previously reported basal to squamous carcinoma transition (BST) phenotype[17,18]. In addition, we uncovered the gene signature enriched in TS3 population compared to other tumor epithelial states (Fig. 1f). While TS3 and BST share hedgehog pathway downregulation (Supplementary Fig. 1d), they have distinct non-overlapping gene expression programs (Fig. 1f, green vs. orange box), suggesting they also possess distinct cellular resistance trajectories. Among the other tumor epithelial states, TS1 and TS2 are basal and undifferentiated in nature, TS4 and BST are tumor epithelial cells that are more differentiated and undergone squamous transition, while cycling and proliferative tumor states represent tumor cells undergoing cell cycling or in active proliferation respectively.

As some of the marker genes for TS3 resemble epithelial to mesenchymal transition (EMT), we sought to investigate to what extent TS3 populations demonstrate an EMT phenotype. Analysis of both epithelial and mesenchymal populations from the same 2 treatment naïve BCC tumors unveiled a separate EMT cluster that transits from epithelial state to mesenchymal state (Supplementary Fig. 1e). The EMT cluster is distinguished from fibroblasts based on its high expression level of Keratin 14 and absence of fibroblast markers PDGFRA or PDGFRB. Meanwhile, the EMT cluster is high in the expression of mesenchymal marker Vimentin and overall expression of EMT signature genes (Supplementary Figs. 1f-g). In comparison, the TS3 signature scores strikingly low as an EMT cluster (Supplementary Fig. 1 g). Consistently, TS3 tumor has higher expression of epithelial CDH1 (coding for E-cadherin) and lower expression of Vimentin compared to the EMT cluster (Supplementary Fig. 1f), suggesting TS3 tumor state is largely epithelial in nature despite upregulation of some genes shared in EMT. Gene ontology (GO) analysis of TS3 signature genes also did not show an enrichment of EMT-related terms (Supplementary Fig. 1h).

To gain insights into TS3 prevalence among treatment naïve BCC tumors, we scored the tumor epithelial populations from 17 human BCC scRNA-Seq datasets (Supplementary Figs. 1i) across different studies[18,20,21] for expression levels of TS3 gene signature and observed heterogeneity in expression levels from patient to patient with roughly 1/3 expressing elevated TS3 signature scores (Fig. 1g). TS3 signature was further shown to be associated with poor overall survival in various solid tumors such as pancreatic adenocarcinoma, head and neck squamous cell carcinoma, mesothelioma and brain lower-grade glioma (Fig. 1h).

### Tumor epithelial basal to inflammatory transition (BIT) is regulated by inflammatory NF-κB family of transcription factors

We further defined the distinct lineage trajectories and transcription factor (TF) network of TS3 and BST. We initially confirmed the distinct TS3 and BST lineage trajectories in naïve human BCCs using Monocle pseudotime analysis[22] by demonstrating the different TS3 and BST endpoints (Fig. 2a, b). Focusing on differential transcriptional

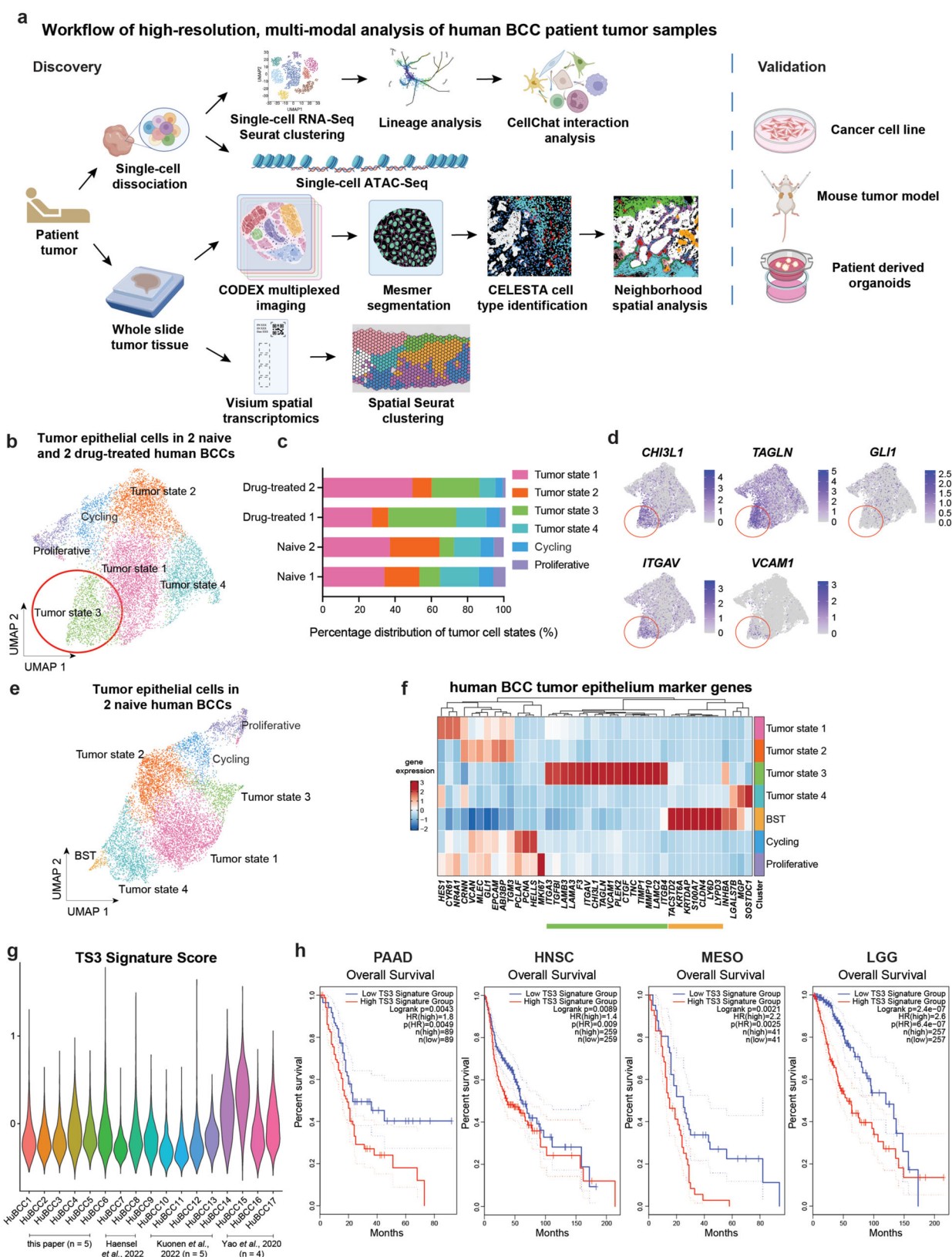

**a** Workflow of high-resolution, multi-modal analysis of human BCC patient tumor samples

**b** Tumor epithelial cells in 2 naive and 2 drug-treated human BCCs

**c** Percentage distribution of tumor cell states (%)

**d** CHI3L1, TAGLN, GLI1, ITGAV, VCAM1

**e** Tumor epithelial cells in 2 naive human BCCs

**f** human BCC tumor epithelium marker genes

**g** TS3 Signature Score

**h** PAAD Overall Survival, HNSC Overall Survival, MESO Overall Survival, LGG Overall Survival

regulators, we performed scATAC-Seq followed by integrated analysis of two human BCCs using ArchR[23] and identified cell types that closely mirrored those identified by scRNA-Seq analysis (Supplementary Figs. 2a–d). Specifically, tumor epithelial clusters C1-C5 were highly analogous to tumor states identified by scRNA-Seq analysis (Fig. 2c, d, Supplementary Fig. 2e), with C1 (orange) equivalent to BST and C3

(green) equivalent to TS3. Consistently, integration of scRNA-Seq data with scATAC-Seq data demonstrated distinct chromatin accessibility profiles at their respective cluster markers between TS3 and BST (Fig. 2e).

While our previous work delineated the central role for AP-1 family of TFs as molecular drivers of BST[17], examination of the TF motifs

**Fig. 1 | Integrated single cell analysis of human BCC tumors identifies a tumor state (TS3) enriched by drug treatment. a** Schematic workflow of high-resolution, multi-modal analysis of human BCC patient tumor samples. **b** UMAP plot of scRNA-Seq data consisting of 11041 cells from two naïve and two drug-treated human BCC tumor epithelial cells with six distinct clusters. Tumor state 3 is circled. **c** Percentage distribution of six tumor cell states in two naïve and two drug-treated human BCC tumor epithelial cells. **d** Feature plots highlighting the expression of key markers (*CHI3L1*, *TAGLN*, *ITGAV* and *VCAM1*) of tumor state 3 as well as expression of hedgehog signaling driver *GLI1*. **e** UMAP plot of scRNA-Seq data consisting of 8840 cells from two naïve human BCC tumor epithelial cells with seven distinct clusters. **f** Heatmap showing the marker genes for the various tumor epithelial clusters from (**e**). Tumor state 3 cluster marker gene signature is highlighted in green while BST cluster marker gene signature is highlighted in orange.

**g** Violin plot of Tumor state 3 (TS3) signature gene scoring across tumor epithelial cells from 17 human BCC patients across different studies[18,20,21]. **h** Survival curves between patients with high and low expression levels of Tumor state 3 (TS3) gene signature. The 95% confidence intervals are shown as dotted lines. PAAD patients: 89 patients in high and low TS3 signature groups; HNSC patients: 259 patients in high and low TS3 signature groups; MESO patients: 41 patients in high and low TS3 signature groups; LGG patients: 257 patients in high and low TS3 signature groups. PAAD: Pancreatic adenocarcinoma; HNSC: Head and Neck squamous cell carcinoma; MESO: Mesothelioma; LGG: Brain Lower Grade Glioma. **a** is created with BioRender.com released under a Creative Commons Attribution-NonCommercial-NoDerivs 4.0 International license. Source data are provided as a Source Data file.

enriched in TS3 (C3) revealed motifs for the inflammation-associated NF-κB family of TFs (Fig. 2f, Supplementary Fig. 2f). Further, expression levels of TF regulators in the NF-κB family (*NFKB1, NFKB2, RELA, RELB*) were highest in TS3 cluster (Fig. 2g, h), reinforcing the notion that NF-κB pathway activation within the tumor epithelium drives TS3. To further confirm the NF-κB signaling activity in TS3 at gene expression levels, we performed gene scoring for genes in the NF-κB signaling pathway among tumor epithelial populations. The elevated NF-κB signaling score was observed in TS3 to a greater degree than in other tumor states of naïve BCCs (Fig. 2i). In addition, we observed an elevated NF-κB signaling score in drug-treated tumor epithelial populations compared to naïve tumor epithelial populations (Fig. 2j), supporting a potential role of elevated NF-κB signaling in mediating therapy resistance. Consistently, NF-κB signaling scores were also highest among BCCs with highest TS3 signature score (Gorlin syndrome BCCs from Yao et al.[20] which are more inflammatory in nature) (Supplementary Fig. 2g). We therefore term the developmental trajectory undertaken by BCC tumor state TS3 as tumor epithelial basal to inflammatory transition (BIT).

## BIT tumor epithelium is localized in a pro-inflammatory niche enriched with myeloid cells

To interrogate the local neighborhood environments that facilitate BIT and BST, we used CO-Detection by indEXing (CODEX) multiplexed tissue imaging[24] to visualize the spatial organization of distinct tumor states in whole slide sections from 4 human BCC tumors. We designed a 41-marker CODEX panel to include specific tumor markers for each of the tumor states based on scRNA-Seq findings (Fig. 3a, Supplementary Data 1 and 2). Markers for epithelium (KRT14), fibroblasts (FAP), macrophages (CD68), vasculature (CD31), T cells (CD3), and neutrophils (MPO) were clearly visualized in the CODEX fluorescent images (Fig. 3b); tumor state specific markers (TROP2 and CHI3L1) further allowed the spatial lineation of BST and BIT subregions of tumors respectively (Fig. 3c).

We used the unsupervised machine learning cell type identification method CELl typE identification with SpaTiAl information (CELESTA)[25] to annotate distinct cell types and tumor states (Fig. 3d, Supplementary Figs. 3a–c). CELESTA identified a total of 17 distinct cell types (Supplementary Figs. 3a–b, Supplementary Data 3). Our analysis corroborated previous work[18] demonstrating that BST tumor epithelium occurs centrally and compactly within tumor nodules and does not interface directly with stromal elements (Fig. 3d). Unlike BST tumor epithelium, BIT tumor epithelium exhibits a highly branched/infiltrative morphology in close proximity to a myeloid-rich neighboring stroma in the superficial aspect of the tumor (Fig. 3d). To obtain a quantitative, unsupervised view of BCC tissue architecture, cellular neighborhood (CN) analysis was performed using k-nearest-neighbors (k-NN) approach[26] to identify a total of 9 spatial communities, where each spatial community was defined by a localized enrichment of specific cell types within the tissue that mediate cellular interactions (Fig. 3e, Supplementary Figs. 3d–e). As controls we confirmed that

fibroblasts dominated the "Stromal", endothelial cells the "Vasculature", and activated T cells the "T cell" enriched communities, respectively. We also confirmed previous work defining a highly immune-reactive region termed Immune Swarm[16], where we observe extensive T cell infiltration in "T cell" communities (cyan) and correspondingly the elimination of tumor epithelium in those subregions.

By contrast our analysis revealed three distinct tumor epithelial-enriched spatial communities: tumor (non-BIT/BST) (white), BST tumor (orange) and BIT tumor (green). Cell-type composition differed among the neighborhoods surrounding identified tumor epithelial spatial communities (Fig. 3f). Surrounding the BST community were mainly other tumors cells, consistent with the observed lack of contact to surrounding stromal elements. By contrast, increased proportions of myeloid and immune cells encircled the BIT epithelium that was not seen with non-BIT/BST tumor epithelium (white, Fig. 3d). While both the tumor (non-BIT/BST) and BIT communities contained similar proportions of fibroblasts and dendritic cells (Fig. 3g), the BIT neighborhood was enriched for inflammation-associated macrophages, neutrophils, vasculature, and immune cells. Further, distance analysis across all samples between the three tumor states and myeloid cells demonstrated striking proximity of BIT tumor cells to macrophages and neutrophils compared to tumor (non-BIT/BST) and BST tumor cells (Fig. 3h, Supplementary Fig. 3f). Overall, our high-resolution, unsupervised analysis of global spatial communities demonstrates that BIT tumor epithelium arises within a pro-inflammatory myeloid-dominant community, spatially and cellularly distinct from the stroma-deplete BST community.

## TREM1 myeloid cell-derived IL1 and OSM signaling correlates with BIT

Our analysis of global spatial communities highlighted a relationship between BIT tumor epithelium and inflammation-associated myeloid cells, but the identity of key druggable microenvironmental inducers facilitating BIT tumor-stromal crosstalk remained elusive. We, therefore, applied CellChat analysis[27] to examine potential cell-cell interactions between tumor epithelial cells and other cell types in the BCC stromal microenvironment using human single-cell gene expression data. We identified the highest numbers of outgoing and incoming signaling interactions in BIT tumor state, TREM1 and TREM2 myeloid cells, and dendritic cells, with fewer interactions predicted from T cells and fibroblasts (Supplementary Figs. 4a–c). We therefore focused our CellChat ligand-receptor pair interaction analysis on tumor epithelial and myeloid subpopulations (Fig. 4a). BIT tumor epithelium possessed higher interaction strength compared to the other tumor epithelial states (Fig. 4b). A screen of signaling patterns predicted strong signaling interactions between TREM1 myeloid cells and BIT tumor epithelium, with candidate TREM1 myeloid cell-derived signals including EGF, VISFATIN, IL1, TGF-β, and OSM (Fig. 4c, d). Of note two signals in particular, IL1 signaling and OSM signaling, demonstrated specificity towards BIT tumor epithelium compared to other tumor epithelial states (Fig. 4e, f, Supplementary Figs. 4d–e).

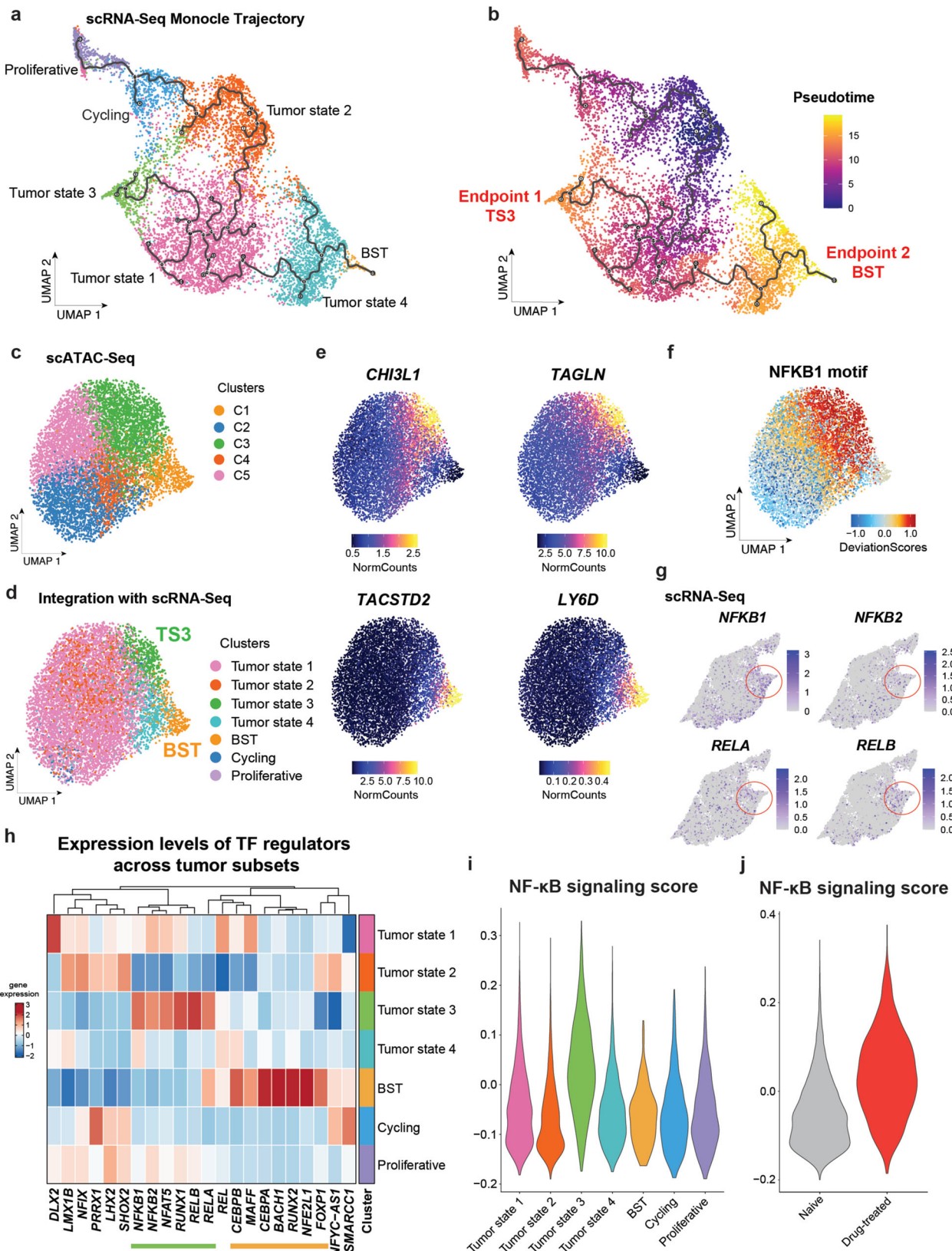

We sought to further strengthen the association between BIT tumor epithelium and TREM1 myeloid cells by interrogating scRNA-Seq data from myeloid populations derived from four naïve human BCCs, two of which had more abundant BIT tumor epithelium and two of which had less abundant BIT tumor epithelium (Supplementary Figs. 4f–g). We reasoned that TREM1, inflammation and IL1/OSM

expression would correlate with abundance of BIT tumor epithelium. Indeed, high-resolution clustering of myeloid cells identified five subpopulations: TREM1 and TREM2 myeloid cells, two dendritic cell clusters and one Langerhans cell cluster (Fig. 4g). Importantly, we observed an expansion in proportion of TREM1 myeloid cells in BCC tumors with more abundant BIT tumor epithelium compared to those

**Fig. 2 | Tumor epithelial basal to inflammatory transition (BIT) is regulated by inflammatory NF-κB family of transcription factors. a** Monocle trajectory projection of 8840 tumor epithelial cells from two naïve human BCC tumors with the same cluster information from (**e**). **b** Pseudotime analysis of 8840 tumor epithelial cells from two naïve human BCC tumors with the same UMAP projection from (**a**). **c** UMAP plot of scATAC-Seq data consisting of 8163 tumor epithelial cells from two naïve human BCC tumors with five distinct clusters. **d** UMAP plot of scRNA-Seq and scATAC-Seq data integration consisting of 8163 tumor epithelial cells from two naïve human BCC tumors. **e** Feature plots of the GeneIntegrationMatrix for *CHI3L1*, *TAGLN*, *TACSTD2* and *LY6D* for scRNA-Seq and scATAC-Seq data integration of tumor epithelial cells from two naïve human BCC tumors as shown in (**d**). **f** Feature plot of the motif DeviationMatrix for NFKB1 motif in scATAC-Seq data with the

same UMAP projection from (**c**). The higher the deviation score indicates greater enrichment of the TF motif. **g** Feature plots highlighting the expression of NF-κB family of transcription factors (*NFKB1*, *NFKB2*, *RELA* and *RELB*) in tumor state 3 (circled) for scRNA-Seq data with the same UMAP projection from (**e**). **h** Heatmap showing the expression of transcription factor (TF) regulators across tumor epithelial states from two naïve human BCC tumors from (**e**). TF regulators highly expressed in Tumor state 3 (TS3) are highlighted in green while TF regulators highly expressed in BST cluster are highlighted in orange. **i** Violin plot of NF-κB signaling signature gene scoring across tumor epithelial cells from two naïve human BCC tumors from (**e**). **j** Violin plot of NF-κB signaling signature gene scoring across tumor epithelial cells from two naïve and two drug-treated human BCC tumors from (**b**).

with less abundant BIT tumor epithelium (Fig. 4h, i). Further, gene scoring on the myeloid cells demonstrated that TREM1 myeloid cells had the highest inflammatory score across all myeloid subpopulations (Fig. 4j). Detailed phenotypic profiling showed that TREM1 myeloid cells were enriched for *IL1A*, *OSM*, and *IL6* expression compared to other myeloid populations (Fig. 4k), further supporting the association between BIT tumor epithelium and TREM1 myeloid-derived IL1A and OSM.

## BIT tumor epithelium arises in a specialized inflammatory environment defined by tumor-associated TREM1 myeloid cell gene signature

While TREM1-expressing myeloid cells have been implicated in a wide variety of both cancer and non-cancer-associated inflammation[28], the spectrum of their effector signals in each context, in particular with respect to cancer drug resistance, remains poorly elucidated. To address this deficit, we created a 16-gene signature depicting TREM1 molecular and ligand secretion profile based on TREM1 myeloid cluster marker genes from four naïve human BCCs, two of which had a higher abundance of BIT tumor epithelial cells and two of which had a lower abundance of BIT tumor epithelial cells (Fig. 5a, Supplementary Fig. 4f, Supplementary Data 4). Expression levels of the genes within the 16-gene TREM1 myeloid signature were elevated in myeloid cells from BIT-high BCCs relative to BIT-low BCCs (Fig. 5a). In contrast, expression of the TREM2 myeloid signature was elevated in BIT-low BCC myeloid cells (Supplementary Figs. 4g, 5a), an observation consistent with our previous work on drug-sensitive BCC epithelium[16]. We then used Visium spatial transcriptomics to investigate the spatial localization of respective myeloid signatures in human BCCs. Unsupervised clustering of Visium datasets identified cell clusters highly analogous to CODEX-identified spatial communities in matched patient samples (Fig. 5b, Supplementary Figs. 5b, c), and spatial gene scoring of BIT and BST marker genes corroborated the same spatial communities (Fig. 5c, Supplementary Fig. 5d). Importantly, unlike the TREM2 myeloid cell signature, the 16-gene BIT-associated TREM1 myeloid signature was specifically enriched in BIT regions spatially (Figs. 5d–f, Supplementary Fig. 5e), reinforcing the spatial association between BIT tumor epithelium and the specialized TREM1 myeloid cell-associated inflammatory environment.

To evaluate the extent to which the 16-gene TREM1 myeloid signature was tumor-specific, we merged the myeloid populations from BCC tumors with existing scRNA-Seq datasets from healthy individuals, individuals with a variety of inflammatory skin diseases, and patients with different types of cancer[29–32]. The 16-gene TREM1 myeloid signature score was low among eczema and psoriasis patients despite the loss of epithelial barrier function associated with those skin conditions, whereas BIT associated TREM1 myeloid signature was higher in cancers that are more closely associated with inflammation (colorectal, ovarian and thyroid cancers) (Fig. 5g). This data supports the association of TREM1 signature with inflammation in a tumor-specific context rather than general inflammation or skin barrier dysfunction

outside of tumor context. We also investigated additional available wound healing datasets to determine if general wound healing promotes BIT. Among diabetic foot ulcer patients[33], we did not observe BIT induction in either healing or non-healing wounds–again supporting the notion that the specialized inflammatory niche is tumor-specific (Supplementary Fig. 5f). Reinforcing the specialized nature of the inflammation, we interrogated whether our mouse BCC scRNA-Seq datasets from our defined *Ptch1^{+/−}*;*p53^{f/f}*;*K14Cre-ER*;*RFP^{f-s-f}* BCC tumor model that we found typically models many aspects of clinical BCCs[34,35] also possessed BIT-associated inflammation. Surprisingly, gene scoring of BIT signature was generally low across mouse tumor epithelial populations (Supplementary Fig. 5g). Lastly, to contextualize the role of TREM1 myeloid signature in other solid tumors in relation to therapy response, we evaluated scRNA-Seq datasets of human metastatic urothelial cancer and human biliary tract cancer undergone immune checkpoint blockade therapy, with a focus on myeloid populations[36,37]. Strikingly, in both cancer types, we observed a statistically significant higher proportion of myeloid cells with high expression levels of TREM1 myeloid signature in the peripheral blood of non-responders compared to responders of immune checkpoint blockade therapy (Supplementary Figs. 5h, i), suggesting the prognostic value of the tumor-associated TREM1 myeloid signature. The same was not observed in BCC patients undergoing immune checkpoint blockade therapy[19] (Supplementary Fig. 5j), suggesting that instead of inducing further myeloid cell recruitment, immune checkpoint blockade likely enrich for the BCC cells that have already undergone BIT transition in the specialized inflammatory milieu prior to therapy. Overall, spatial transcriptomic data corroborates with and complements our CODEX data, strengthening the spatial association between BIT tumor epithelium and a specialized inflammatory environment defined by TREM1 myeloid cells.

## Il1 and Osm combination treatment induces BIT in BCC cells in vitro via NF-κB

We capitalized on the observation that our mouse BCC model failed to demonstrate the BIT associated tumor epithelium to test the sufficiency of Trem1 myeloid cell-derived ligands in inducing tumor epithelial BIT. We used a mouse BCC cell line ASZ-001 to screen an array of ligands by adding recombinant ligand proteins to the culture media separately or in combination. As with the mouse tumors, BCC cell lines derived from the tumor had undetectable levels of BIT marker *Chil1* (Fig. 6a). Strikingly, the combination of Il1 and Osm ligands dramatically induced *Chil1*, in a synergistic fashion within 24 hours of exposure to ligands (Fig. 6a). Combination of Il1 and Il6 ligands also induced *Chil1* expression to a moderate level (Fig. 6a). This was in sharp contrast to the other ligands screened (Ifn-γ, Tweak and Tgf-β) that failed to induce *Chil1* expression, supporting the sufficiency of Il1 and Osm in converting BCC cells from basal state to BIT tumor state. Principal component analysis (PCA) of RNA-Seq of untreated and 24 and 48 h treated mouse BCC cells corroborated the synergistic activity of Il1 and Osm, whereby combination of the two ligands drove the majority of the

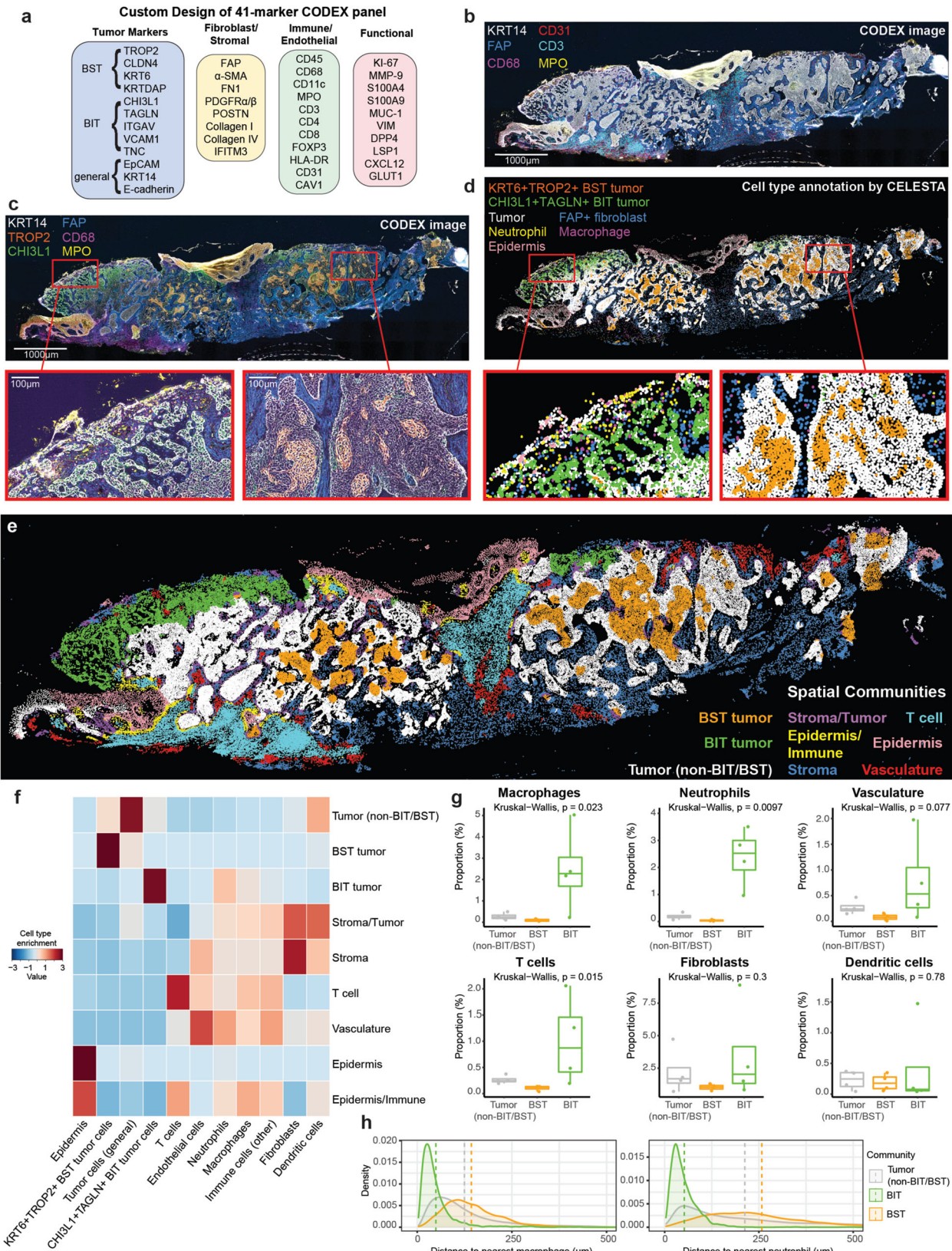

variance seen in transcriptomic data of BCC cells (Fig. 6b). As predicted, the top differentially upregulated genes in BCC cells upon 48 h of Il1a and Osm combination treatment included BIT-associated proinflammatory cytokines and cell-cell communication molecules such as acute-phase *Saa* genes, *Chil1*, *Orm1*, *Orm2* and *Cxcl5* (Fig. 6c). Gene Ontology (GO) analysis revealed that the top 100 significantly

upregulated genes were involved in pathways such as acute inflammatory response, chemokine and cytokine mediated signaling pathway, JAK-STAT and interleukin-1 signaling (Fig. 6d, Supplementary Data 5).

Moreover, gene set enrichment analysis (GSEA) demonstrated a significant enrichment in inflammatory response genes accompanied by a downregulation of hallmark hedgehog signature genes (Fig. 6e).

**Fig. 3 | BIT tumor epithelium is localized in a pro-inflammatory niche enriched with myeloid cells. a** CODEX antibody panel consisting of 41 protein markers of interest. **b** Representative whole slide CODEX image of human BCC tumor depicted as a six-color overlay image (*n* = 4 CODEX samples). Scale bar, 1000 μm. **c** Representative whole slide CODEX image and higher-magnification six-color overlay images of human BCC tumor (*n* = 4 CODEX samples). Scale bar, 1000 μm (whole slide image) and 100 μm (higher-magnification images). **d** Representative whole slide image and higher-magnification images depicting cell types annotated by CELESTA. **e** Identification of nine distinct spatial communities based on CELESTA-assigned cell types and their respective frequencies within each spatial community. **f** Heatmap depicting cell type enrichment of CELESTA-assigned cell types within each spatial community. **g** Absolute proportions (%) of macrophages, neutrophils, vasculature, T cells, fibroblasts, and dendritic cells in each of the three tumor spatial communities: Tumor (non-BIT/BST), BST and BIT. The center corresponds to the median value. The lower and upper hinges correspond to the 25th and 75th percentiles. The upper whisker extends from the hinge to the largest value no further than 1.5*IQR (Inter-Quartile Range). The lower whisker extends from lower hinge to the smallest value at most 1.5*IQR. *P*-value calculated using nonparametric Kruskal-Wallis test for multiple groups. The statistical comparisons were performed across all four samples included in the CODEX spatial analysis. **h** Density plots depicting distributions of the distances from the three tumor types: tumor (non-BIT/BST), BST, and BIT, to nearest macrophages and neutrophils across all four samples. Vertical dotted lines represent the mean distance for each tumor community. Source data are provided as a Source Data file.

Specifically, we observed downregulation of both neuropilin 1 (*Nrp1*) and neuropilin 2 (*Nrp2*), which are known to play important roles as positive regulators of Hedgehog signal transduction[38] (Supplementary Fig. 6a). In addition, Gai family of GPCR heterotrimeric G-proteins (encoded by *Gnai1*, *Gnai2* and *Gnaz* genes) were downregulated upon ligand treatment, which may have contributed to additional suppression of Hedgehog signaling through enhanced protein kinase A activity[39,40].

We performed ATAC-Seq to investigate ligand-induced BIT at the chromatin accessibility level and found 11,531 differentially open chromatin sites and 11,749 differentially closed chromatin sites at the genomic level (Supplementary Figs. 6b-c). Specifically, we observed increased chromatin accessibility at BIT marker gene loci *Chil1* and *Vcam1* and decreased chromatin accessibility at BST marker gene loci *Tacstd2*, *Cldn4* and *Lypd3* upon Il1a and Osm treatment, supporting the induction of BIT phenotype by those ligands (Supplementary Fig. 6d). Motif enrichment analysis of the differentially open chromatin sites at the promoters of upregulated genes revealed significant enrichment of NF-κB family TF motifs (Fig. 6f). We then confirmed the genomic occupancy of NFKB1 using CUT&RUN assay[41] (Supplementary Figs. 6e-f). Overlapping NFKB1 DNA-binding sites with differentially open chromatin sites allowed the identification of 697 high-confidence NFKB1 binding sites and 567 associated genes upon Il1a and Osm combination treatment (Supplementary Fig. 6g, Supplementary Data 6). In addition, genes associated with NFKB1 DNA-binding sites were upregulated at the transcriptional level upon ligand treatment (Supplementary Fig. 6h, Supplementary Data 7). Specifically, upon Il1a and Osm combination treatment for 48 h, we observed enhanced DNA-binding by NFKB1 at the intergenic genomic locus slightly upstream of *Chil1* (Fig. 6g), along with increased chromatin accessibility at *Chil1* promoter locus and enhanced *Chil1* expression (Fig. 6h, Supplementary Fig. 6h), supporting NFKB1's role as a direct regulator of BIT marker gene *Chil1*. To investigate whether BIT phenotype is reversible by drugs and/or cytokine withdrawal, we evaluated *Chil1* expression level upon 24 h of Il1a and Osm induction, followed by 48 h of cytokine withdrawal or Il1 receptor antagonist. Notably, *Chil1* expression restored to baseline levels following cytokine withdrawal or Il1 receptor antagonist treatment (Fig. 6i), indicating the reversible nature of BIT and the potential to therapeutically target the transition. Consistently, pretreatment of BCC cells with NFKB inhibitors JSH23 or withaferin A (WFA), or STAT3 inhibitor ruxolitinib was able to rescue BIT induction by Il1a and Osm (Fig. 6j), demonstrating functional involvements of NFKB1 and STAT3 in BIT induction. Overall, our in vitro approaches demonstrated the synergistic activity of Il1 and Osm in BIT induction; our multiomic investigation further supported the direct involvement of NF-κB in ligand-induced BIT.

### IL1 and OSM combination treatment induces BIT in vivo and ex vivo, leading to resistance to SMO[i] therapy
To investigate the role of Il1 and Osm in BIT induction in vivo, we generated allograft secondary BCC tumors in Nod-Scid mice using primary mouse BCC tumors from our X-ray irradiated *Ptch1*[+/−]*;p53*[f/f]*;K14Cre-ER;RFP*[f-s-f] mouse model, followed by intra-tumoral injections of PBS control or Il1a + Osm ligand cocktail at 3-day interval for 18 days (Fig. 7a). Consistent with our in vitro findings, we observed a significant induction of *Chil1* expression in ligand-treated tumors compared to tumors treated with vehicle (Fig. 7a, b), demonstrating the sufficiency of Il1a and Osm combination treatment in BIT induction in vivo. Similarly, we demonstrated the sufficiency of Il1a and Osm combination treatment in *Chil1* induction in mouse primary BCC tumor-derived organoids ex vivo (Fig. 7c). Transcriptomic profiling of mouse primary BCC tumor-derived organoids following Il1a and Osm combination treatment demonstrated genome-wide changes consistent with those observed in mouse BCC cell line (Supplementary Figs. 7a, b). To evaluate how ligand-induced BIT affects SMO[i] sensitivity, we designed an ex vivo BIT induction scheme using our human patient-derived organoid models, where primary human BCC tumor tissues were embedded as small pieces in basement membrane extract and grown in organoid medium for 7 days with or without IL1A and OSM combination treatment (Fig. 7d). Patient-derived organoids were subsequently treated with SMO[i] vismodegib for 48 h and assayed for sensitivity using Apotracker Green as readouts. Strikingly, we observed significant reduction in the proportion of ApoGreen[+] (SMO[i] sensitive) epithelial cells upon IL1A and OSM combination treatment in five out of the six human BCCs evaluated (Fig. 7e–g, Supplementary Fig. 7c), confirming the clinically relevant roles of IL1A and OSM in mediating SMO[i] resistance.

In summary, our study identifies BIT as a SMO[i] resistant tumor epithelial cell state defined by distinct markers such as *CHI3L1*. In contrast to stromal-independent BST tumor epithelium, BIT tumor epithelium is spatially localized in a distinct pro-inflammatory niche near the epidermis and actively crosstalks with proximal TREM1 myeloid cells. Specifically, tumor-associated TREM1 myeloid cells define the specialized inflammatory environment in which BIT tumor epithelium arises. IL1 and OSM secreted by TREM1 myeloid cells activate the inflammatory NF-κB family of transcription factors within the BIT tumor epithelium, leading to transcriptional upregulation of BIT markers (Fig. 7h). Our work provides critical insights into the BIT resistant state as a targetable tumor state driven by a specialized inflammatory microenvironment.

## Discussion
Our work identifies BIT as a spatially-localized NF-κB-driven tumor epithelial resistance state associated with TREM1-myeloid dominated inflammation characterized by high levels of cytokines IL1 and OSM. BIT clinical importance stems from its existence within many epithelial tumors that correlate with poor patient survival. Our results deepen our understanding of the local tumor microenvironment, define a common sub-type of pro-tumorigenic cancer-associated inflammatory response, and nominate candidate drug targets to counteract it.

Our spatial expression studies provide a detailed analysis of dynamic tumor plasticity in naïve tumors that gives rise to both drug-sensitive and drug-resistant populations within the same BCC tumor.

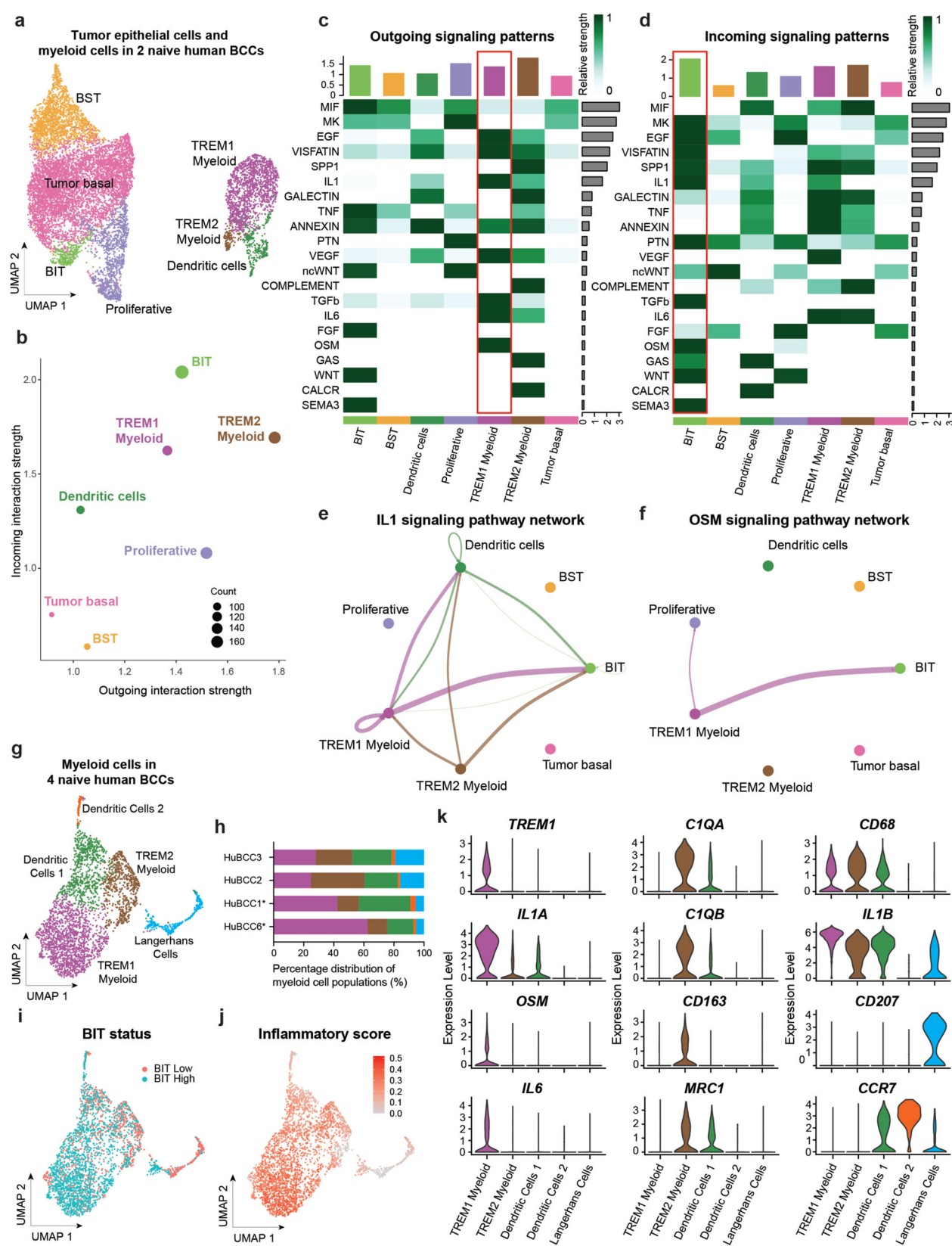

Sensitive, sonic hedgehog (Shh) pathway-driven BCC epithelium arises through the ability to recruit and instruct TREM2 + VCAM1+ myeloid cells into a self-propagating phenotype that allows mutual proliferation and tumor growth with a dearth of effector lymphocytes present in the local environment[16]. Rising within BCC epithelium, BST intrinsically occurs as tumor epithelia become separated from the epithelial

basement membrane and surrounding microenvironment. By contrast, Shh pathway-driven BCC epithelia growing within a specialized TREM1 myeloid-driven inflammatory environment are induced by the TME to undergo phenotypic switching towards the alternative BIT tumor state that possesses low hedgehog pathway signaling and distinct resistance characteristics, providing multiple avenues to thwart

**Fig. 4 | TREM1 myeloid cell-derived IL1 and OSM signaling correlates with BIT.**
**a** UMAP plot of tumor epithelium and myeloid populations consisting of 11375 cells for CellChat analysis (*n* = 2 naïve human BCC tumors). **b** Scatter plot depicting the dominant senders and receivers of outgoing and incoming interactions respectively in a 2D space based on signaling role analysis on the aggregated cell-cell communication network from all signaling pathways from data in (**a**). **c** Heatmap depicting signals contributing most to outgoing signaling of each cell group based on signaling role analysis on the aggregated cell-cell communication network from all signaling pathways. **d** Heatmap depicting signals contributing most to incoming signaling of each cell group based on signaling role analysis on the aggregated cell-cell communication network from all signaling pathways. **e** IL1 signaling pathway network from myeloid cells to BIT tumor epithelium. **f** OSM signaling pathway

network from TREM1 myeloid cells to BIT and proliferative tumor epithelium. **g** UMAP plot of scRNA-Seq data of myeloid cell populations consisting of 3144 cells from four naïve human BCC tumors with five distinct clusters. **h** Percentage distribution of five myeloid cell populations across four naïve human BCC tumors in (**g**). Asterisk(*) indicates patient samples that were also analyzed in Figs. 1–3. **i** UMAP plot of scRNA-Seq data of myeloid cell populations consisting of 3144 cells from four naïve human BCC tumors in (**g**) colored by abundance of BIT tumor epithelium. **j** Feature plot of inflammatory signature gene scoring across myeloid cell populations from four naïve human BCC tumors in (**g**). **k** Stacked violin plot of expression levels of various key myeloid-associated markers for each of the five myeloid clusters from (**g**). Source data are provided as a Source Data file.

targeted therapy. Quantitative CODEX spatial analysis indicates that the proportions of tumor cells involved in BIT transition (reflected through neighborhood sizes) vary from patient to patient, along with the characteristic spatial orchestration of myeloid cell types within each BIT neighborhood, suggesting that BIT is contingent upon inherent genetic susceptibility and exposure to proximal micro-environmental factors that induce it, rather than a hard-wired tumor intrinsic cellular trajectory.

Our identification of tumor BIT through CODEX analysis was unexpected for several reasons. First, resistant tumor subtypes have typically been associated with invasion deep into the tissue[42,43]. The presence of normal skin adjacent to our tumor samples allowed us to orient BIT to demonstrate that it arises in the highly inflammatory superficial wound environment, rather than being deeply invasive. BIT tumors are spatially enriched in the superficial aspect of BCC tumors, maintaining tight cell-cell contact and cell adhesion through integrins, inconsistent with an EMT phenotype where cells lose cell-cell contact and migrate/infiltrative into invasive fronts deep in the tumor. Previous studies have associated some of the surface markers with the deeply invasive activin-expressing tumor[21], while our data suggests that BIT arises independent of high activin expression, highlighting the distinctions between invasiveness and drug sensitivity.

Secondly, while skin epithelium differentiation through NF-κB/IL1 signaling has been previously described and occurs commonly[44,45], NF-κB-driven BIT within tumors charts a unique trajectory with clinical prognostic value. Only a fraction of all BCCs examined contain BIT neighborhoods and we did not detect BIT in mouse BCCs with baseline NF-κB signaling until we increased the local concentrations of IL1/OSM in vitro and in vivo. In addition, skin epithelia are well known to express IL1 and auto-induce NF-κB signaling in skin inflammatory conditions such as eczema or psoriasis. However, we show that tumor epithelial growth within the BIT specialized inflammatory microenvironment fuels its viability potential when under selective drug therapy pressure, distinguishing skin disease physiology from supranormal carcinoma-associated signaling.

A surprising finding in our quantitative spatial analysis of naïve BCC tumors was the adjacency of inflammatory neighborhoods that conferred unique immune cell composition and resistance profiles. "T cell" neighborhoods enriched with effector T cells and mature macrophages that lacked tumor epithelium lie immediately adjacent to drug-sensitive proliferating BCC surrounded by TREM2 + VCAM1+ myeloid cells or resistant BIT tumors surrounded by lymphocyte-poor TREM1+ myeloid-driven inflammation. Our analysis suggests that the particular myeloid sub-type (whether mature macrophages/dendritic cells, TREM2 myeloid or TREM1 myeloid) proximal to the BCC epithelium appears to define the adjacent tumor epithelial state, characteristics of the spatial neighborhood and ultimately tumor susceptibility to treatment. Extensive research on monocyte differentiation reveals enormous plasticity of these cells in response to growth factor, metabolic and microbiome constituents[46,47]. Our spatial

analysis indicates that distinct monocyte differentiation trajectories occur prior to and after treatment to create the observed tumor neighborhoods, highlighting the need for more detailed mechanistic studies to control local tumor-myeloid dynamics. Further, our spatial neighborhood analysis highlights the heterogeneity of T cell infiltration in distinct subregions of the same tumor that have different inflammatory profiles, suggesting spatially heterogeneous patterns of immune cell infiltration and the need for holistic re-evaluation of spatial TME profiles beyond the current classification of inflamed (hot) and non-inflamed (cold) tumors.

Our spatial neighborhood analysis describes the full range of tumor-immune dynamics within a naïve epithelial carcinoma, from anti-tumor immune storm to pro-tumor TREM1-associated BIT and stroma-independent BST. While we commonly observe TREM2-associated proliferative BCC and BST in human and murine BCCs, the variable abundance of BIT cells in BCC patients and murine tumors argues for a specialized role of the TREM1-associated inflammation in driving high levels of NF-κB signaling in tumor epithelium. This auto-inflammatory positive feedback loop is driven by elevated levels of IL1 and OSM which are sufficient to induce BIT in murine BCC allografts and are reminiscent of autoinflammatory skin disorders like pustular psoriasis that are associated with neutrophilic infiltrates[48,49]. In line with our findings, IL1 and OSM have been implicated in cancer-associated inflammation and therapy resistance and research has shown that blocking IL1 and IL6 family of inflammatory cytokines could enhance anti-tumor activity in head and neck squamous cell carcinomas[50–52]. Intriguingly, recent single-cell datasets reveal overlaps between TREM1-associated inflammation and myelosuppressive neutrophils[53], a finding we corroborate by establishing the local enrichment of neutrophils in the BIT neighborhood. Future investigations will focus on factors inducing TREM1-associated inflammation and the role of myelosuppressive neutrophils in BCC therapy response.

While our research has provided valuable insights, one significant limitation of our study is the small sample size, making it difficult to draw broad conclusions. Future studies with larger sample sizes are necessary to validate our findings and ensure their robustness. Another limitation pertains to the differences in the inflammatory responses between mouse models and human basal cell carcinoma (BCC). Our findings indicate that while murine models provide a useful platform for functional investigations, they may not entirely mimic the human inflammatory processes involved in BIT. Our ongoing research aims to bridge this gap, enhancing the relevance and applicability of our murine model system to human BCC.

Overall, the identification of TREM1+ myeloid-derived IL1 and OSM as sufficient to induce BIT and resistance generates a unique therapeutic opportunity to target BIT using currently available agents such as IL1R antagonist Anakinra or JAK inhibitors. Our data highlights the need to distinguish pro-tumor inflammation from anti-tumor inflammation and supports the intriguing hypothesis that targeting myeloid-driven BIT-associated inflammation would improve

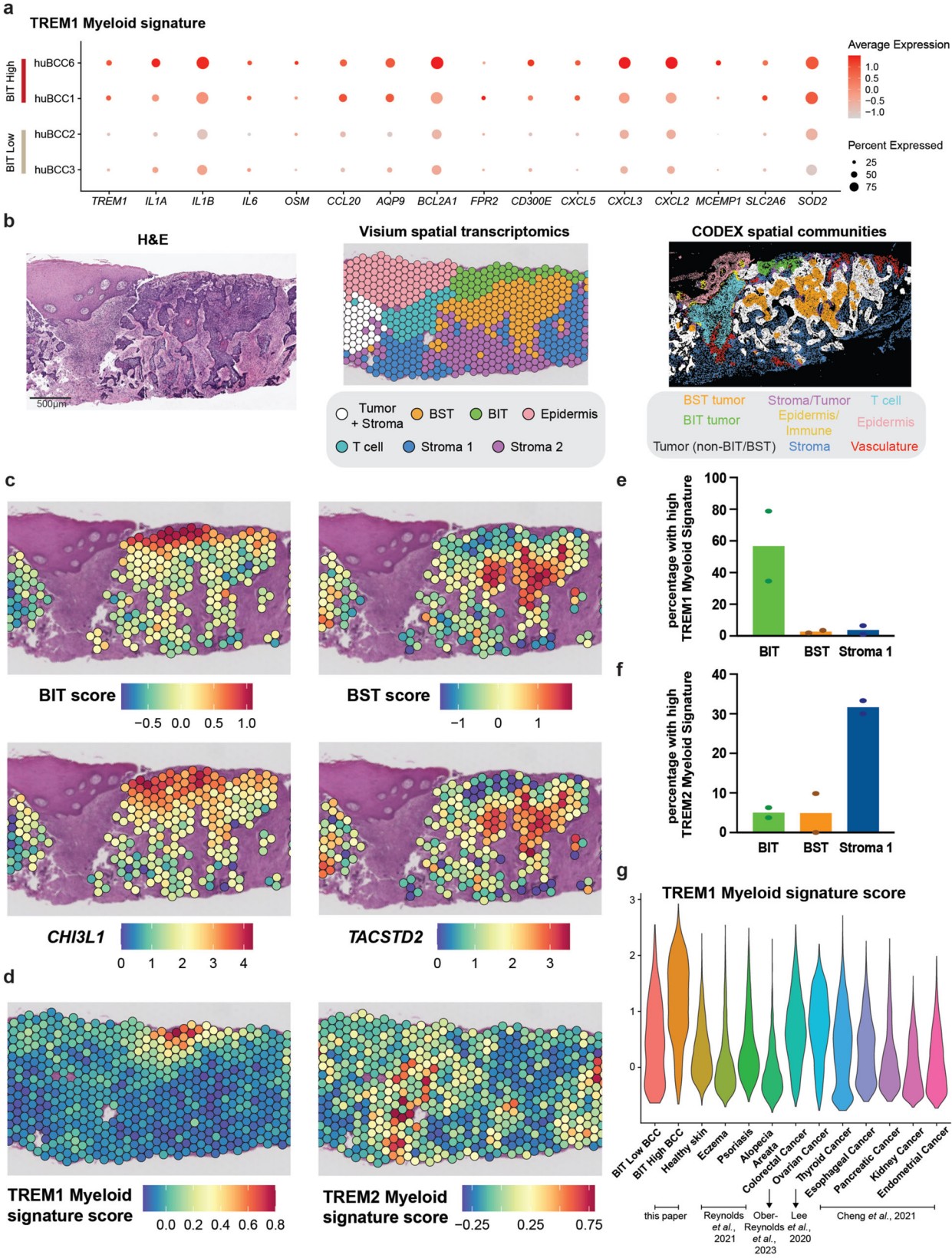

therapeutic outcome without thwarting lymphocyte-driven anti-tumor immune reactions. Agents such as IL1R antagonist, Anakinra, NFKB inhibitors or JAK inhibitors have the potential to be effective partners as part of a combinatorial cocktail that addresses BIT-associated therapy resistance.

## Methods

### Animals

Mice were housed under standard conditions, and animal care was compliant with the protocols approved by the Institutional Animal Care and Use Committee (IACUC) at Stanford University. *Ptch1⁺/⁻;p53ᶠ/ᶠ;*

**Fig. 5 | BIT tumor epithelium arises in a specialized inflammatory environment defined by tumor-associated TREM1 myeloid cell gene signature. a** Dot plot depicting the expression levels of TREM1 myeloid cluster markers in the 16-gene TREM1 myeloid signature across myeloid cell populations in (**g**) stratified by abundance of BIT tumor epithelium in four naïve human BCC tumors.
**b** Hematoxylin and Eosin (H&E) staining image (*left*), spatial DimPlot of Visium spatial Seurat clusters (*middle*) and CODEX spatial communities (*right*) of the same human BCC tumor (*n* = 2 patients). Scale bar = 500 μm. **c** Spatial Seurat gene signature scoring for BIT signature (*top left*) and BST signature (*top right*) on human BCC tumor. Spatial feature plots of *CHI3L1* (*bottom left*) and *TACSTD2* (*bottom right*) for human BCC tumor. **d** Spatial Seurat gene signature scoring for TREM1 myeloid signature and TREM2 myeloid signature on human BCC tumor. **e** Quantification of

percentage of spots with high TREM1 myeloid signature score (top 10 percentile of signature score in Fig. 5d) in each of the three spatial Seurat clusters: BIT, BST and Stroma 1. Height of the bar represents the average percentage in each spatial Seurat cluster (*n* = 2 patients). **f** Quantification of percentage of spots with high TREM2 myeloid signature score (top 10 percentile of signature score in Fig. 5d) in each of the three spatial Seurat clusters: BIT, BST and Stroma 1. Height of the bar represents the average percentage in each spatial Seurat cluster (*n* = 2 patients). **g** Violin plot of TREM1 myeloid signature gene scoring across myeloid cells from naïve human BCC tumors, healthy skin tissue[29], eczema skin tissue[29], psoriatic skin tissue[29], alopecia areata scalp tissue[30], colorectal cancer[32] and many other different types of cancer[31]. Source data are provided as a Source Data file.

*K14Cre-ER* mice (with or without *RFP^{f-s-f}*) were generated and utilized to develop BCC tumors as described previously[34,35]. Cre-mediated recombination was induced through three 4-hydroxytamoxifen injections (50 μl of 5 mg/ml; Sigma;H7904) completed once per day for three consecutive days. Within one day of the final dose, mice were irradiated (5.25 Gy) using an X-ray irradiator. Primary tumors developed four to six months afterward. Irradiated mice were monitored multiple times per week for signs of discomfort and distress and for tumor development. Animals persistently scratching and causing ulcerations despite analgesics and topical antibiotics were considered for euthanasia. Adult female Nod-Scid mice (7–8 weeks of age) (Jackson Laboratory, Stock number: 001303) were used as host animals for grafted tumors. We only used female mice for allografts because male mice bite the grafts. Host mice with grafted tumor cells were monitored at least 5 days a week for signs of discomfort and distress and any mice that exhibited poor grooming and health were euthanized. Tumor graft sites were closely monitored to ensure tumor burden did not exceed the maximal tumor burden at 10% of pre-inoculation body weight of each mouse. Palpable tumors were developed after 1–2 months of grafting, upon which host mice were euthanized using $CO_2$ and tumors were collected for analysis. All experiments were compliant with the ethics guideline.

### Human Samples
All patient samples were obtained through written informed consent and subsequently de-identified. All protocols for sample acquisition and usage are in accordance with the reviewed protocol by the Stanford University Institutional Review Board, protocol #18325 (Stanford, CA).

### Sample processing
Human tumor specimens were briefly rinsed with PBS before being minced to pieces less than 1 mm in diameter. After mincing, tumor pieces were transferred to 50-ml conical tube and 40 ml of 0.5% collagenase solution (Gibco; 17-100-017) in DKFSM media (Gibco; 10744-019). The minced tumors were then incubated at 37 °C with rotation for 2 h. At the end of the 2-hour incubation, 5 ml of 0.25% Trypsin (Gibco; 25200056) was then added to the 50-ml conical tube, which was then further incubated at 37 °C with rotation for an additional 15 mins. 5 ml of FBS was then added to the cellular suspension. Mouse tumor (mBCC) specimens were processed in the same manner except for the initial incubation with collagenase, which was only incubated for 1 h. The dissociated cellular suspensions were filtered with a 70 μm filter before being pelleted. Cells were either immediately used for downstream experiments or frozen down in BAMBANKER™ freezing medium (FujiFilm; CS-02-001).

### Cell lines
The mouse basal cell carcinoma cell line ASZ001 was generated by the Epstein group[54]. HEK293-mNoggin-Fc cells were a gift from the Hans Clevers group[55]. HEK293-HA-R-Spondin1-Fc (R&D Systems; 3710-001-01) and L-WRN cells (ATCC; CRL-3276) were commercially purchased.

ASZ001 were authenticated previously by the Epstein group[54]. In our experiments, ASZ001 cells were monitored by their morphology, sequencing, and utilization of mouse specific primers. In our experiments, HEK293-mNoggin-Fc, HEK293-HA-R-Spondin1-Fc, and L-WRN cells were monitored by their morphology.

ASZ-001 cells (mouse BCC cell line) were grown in M154CF (Thermo Fisher Scientific, Waltham, MA) supplemented with 2% chelated FBS, 1% Penicillin-Streptomycin and 0.05 mM $CaCl_2$[54]. Experiments were carried out in serum-starved conditions. Cell lines were assessed for mycoplasma using the MycoSeq Mycoplasma real-time PCR detection kit (ThermoFisher Scientific, Waltham, MA).

### Real-time quantitative PCR
RNA was isolated from adherent cells using TRIzol Reagent (Invitrogen; 15596026) followed by RNeasy standard protocol (Qiagen) as per manufacturer's instructions. For gene expression analysis in organoids, tissues were suspended in RLT buffer (Qiagen) and homogenized using 2 ml tissue lysing matrix E tubes (MP Biomedicals) and RNA was subsequently extracted using the RNeasy standard protocol (Qiagen). Quantitative real-time PCR was performed using TaqMan assay primers obtained from Thermo Fisher Scientific (Waltham, MA). Each reaction was performed in replicate and values were normalized to Gapdh. The MxPro software was used for data acquisition.

### CODEX Antibody Conjugation and Validation
DNA oligonucleotide barcodes and antibody conjugation kits were obtained from Akoya Biosciences. For conjugations, 50 μg of purified, carrier-free antibodies were used. Antibody clones and barcode information are provided in Supplementary Data 1. Centrifugation was performed at 12,000 *g* for 8 mins for all steps unless otherwise stated. Antibodies were concentrated on a 50 kDa filter equilibrated with filtration buffer. The sulfhydryl groups were activated by incubating for 30 mins at room temperature (RT) with the reduction mix. The antibody was then washed with conjugation buffer once. The oligonucleotide barcode was resuspended in the conjugation buffer and added to the antibody and allowed to incubate for 2 h at RT. The conjugated antibody was washed 3 times, by resuspending and spinning down with purification solution. Antibody was then eluted by adding 100 μl storage buffer and spinning at 3000 *g* for 4 mins. The conjugated antibodies were stored at 4 °C till use.

Before use in the multiplex CODEX experiments, conjugated antibodies were validated with CODEX single-marker stains on tissue derived from patients with basal cell carcinoma. Staining was performed with each conjugated antibody as described below. Screening buffer was prepared according to the CODEX manual provided by Akoya Biosciences. The fixed and stained tissue was incubated in the screening buffer for up to 15 mins. Fluorescent DNA probe was prepared and added to the stained tissue for 5 mins. The tissue was washed 3 times with screening buffer followed by 1X CODEX buffer. The tissue was then imaged using a Keyence BZ-X810 inverted fluorescent microscope. The staining pattern was confirmed by comparing to immunofluorescent images acquired pre-conjugation. Antibodies

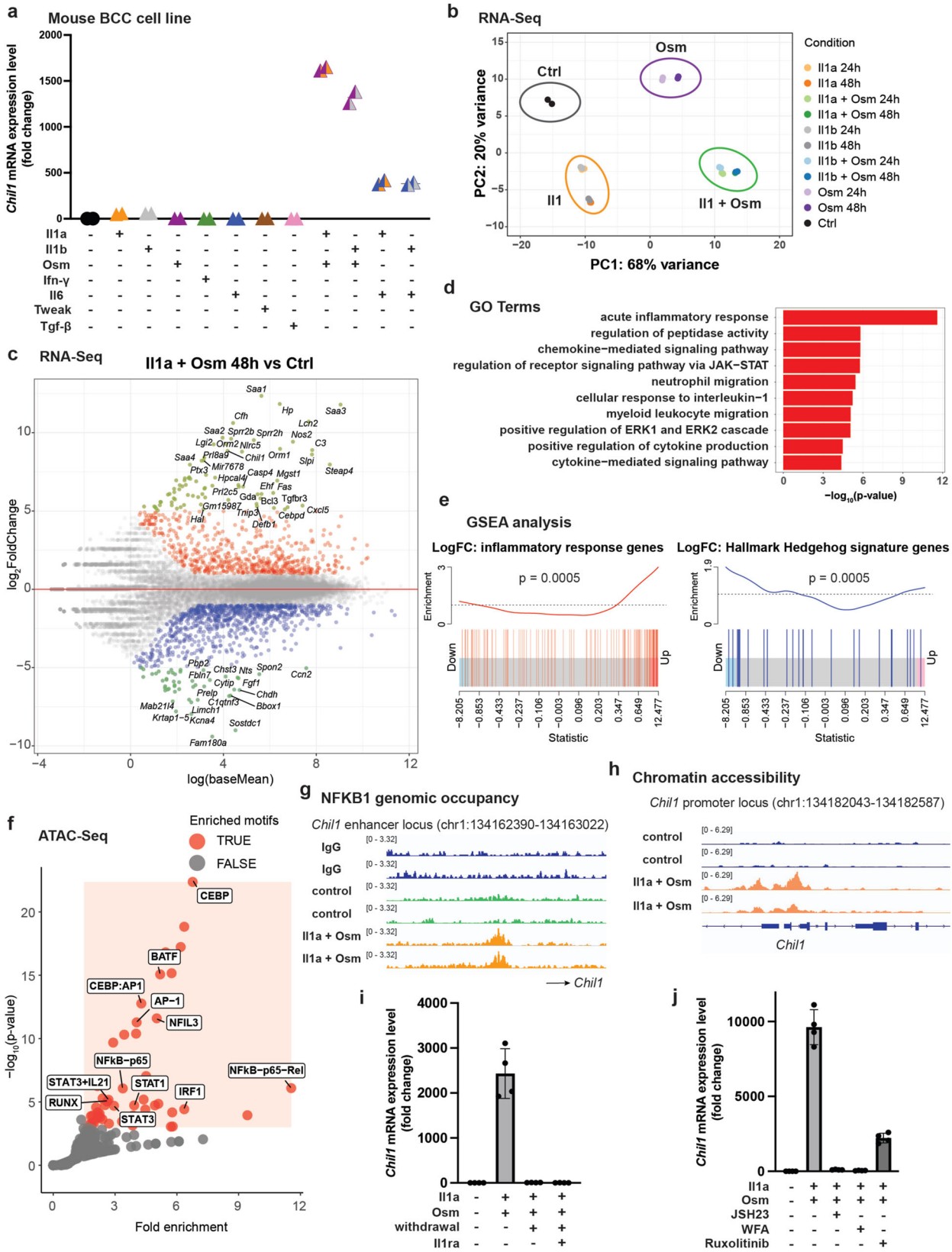

that did not show correct staining pattern were discarded and replaced with clones that displayed specific staining patterns.

**CODEX antibody staining of FFPE tissues**
Deparaffinization was performed using a slide warmer at 65 °C for 30 minutes. The coverslips were allowed to come to room temperature for 5 mins before immersion in xylene for 5 mins twice. Tissue was rehydrated by immersion in descending ethanol concentrations (100% twice, 90% once, 70% once, 50% once, 30% once), followed by two ddH$_2$O washes, for 5 mins each. A two-step antigen retrieval was performed using a citrate-based antigen retrieval solution (NovusBio) followed by 1X Dako antigen retrieval solution (Agilent cat# S2367).

**Fig. 6 | Il1 and Osm combination treatment induces BIT in BCC cells in vitro via NF-κB. a** *Chil1* expression fold change upon 24-h treatment with various recombinant ligand proteins compared to PBS control in ASZ-001 ($n = 2$ biological replicates). **b** Principal-component analysis (PCA) of bulk RNA-seq data obtained from ASZ-001 upon treatment with various recombinant ligand proteins at 24-h and 48-h timepoints compared to PBS control ($n = 2$ biological replicates). **c** Scatterplot of genes with differential expression upon 48-h Il1a + Osm treatment against control ($n = 2$ biological replicates). Genes with expression fold change > 1 and $p < 0.05$ are marked in red, genes with expression fold change < −1 and $p < 0.05$ are marked in blue. Genes with expression fold change > 5 and $p < 0.05$ or fold change < −5 and $p < 0.05$ are marked in green and labeled with gene names. A total of 25120 genes were included in the differential gene expression analysis. Two-sided Wald test was used to compute *p*-value statistics between the control and treatment groups. *P*-values were corrected for multiple testing using Benjamini and Hochberg method. **d** Gene Ontology (GO) Terms associated with the top 100 upregulated genes upon 48-h Il1a + Osm treatment against control. A total of 26994 biological process terms were included in the GO Terms analysis. One-sided Fisher's exact test was used to compute *p*-value statistics for each GO Term. *P*-values were corrected for multiple testing using Benjamini and Hochberg method. **e** Gene set enrichment analysis (GSEA) plots of inflammatory response gene signature (red) and Hallmark Hedgehog gene signature (blue) in RNA-Seq data obtained from ASZ-001 upon 48-h Il1a + Osm treatment against control ($n = 2$ biological replicates). A total of 13522 genes were included in the GSEA analysis. *P*-values were calculated using one-sided

ROAST gene set testing. *P*-values were corrected for multiple testing using Benjamini and Hochberg method. **f** Motif enrichment analysis on ATAC-Seq peaks with increased chromatin accessibility at the promoters of upregulated genes upon 48-h Il1a + Osm treatment against control ($n = 2$ biological replicates). Significantly enriched motifs (fold enrichment ≥ 1.5 and $-\log_{10}$(p-value) ≥ 3) are highlighted in red. A total of 436 known motifs from HOMER Motif Database were included in the motif analysis. *P*-values were calculated using HOMER analysis pipeline. *P*-values were corrected for multiple testing using Benjamini and Hochberg method. **g** Example tracks of NFKB1 genomic occupancy sites near the *Chil1* locus from CUT&RUN sequencing data obtained from ASZ-001 upon 48-hour Il1a + Osm treatment against IgG and PBS controls ($n = 2$ biological replicates). **h** Example tracks of accessible chromatin sites at the *Chil1* locus from ATAC-Seq data obtained from ASZ-001 upon 48-hour Il1a + Osm treatment against control ($n = 2$ biological replicates). **i** *Chil1* expression fold change upon 24 h Il1 + Osm ligand cocktail treatment, 24 h Il1 + Osm ligand cocktail treatment followed by cytokine withdrawal for 48 h, and 24 h Il1 + Osm ligand cocktail treatment followed by cytokine withdrawal and Il1 receptor antagonist treatment for 48 h respectively, compared to PBS control in ASZ-001 ($n = 4$ biological replicates). Data are presented as mean values +/- SD. **j** *Chil1* expression fold change upon 24 h Il1 + Osm ligand cocktail treatment with or without 6 h pretreatment with NFKB inhibitors JSH23 or withaferin A (WFA) or JAK/STAT3 inhibitor ruxolitinib respectively in ASZ-001 ($n = 4$ biological replicates). Data are presented as mean values +/- SD. Source data are provided as a Source Data file.

Antigen retrieval was performed by first heating in a pressure cooker for 20 mins in 1X citrate buffer, pH 6.0 followed by heating for 20 mins at high pressure in 1X Dako antigen retrieval buffer. The coverslips were allowed to come to RT and then washed twice in ddH₂O after each heating step. Autofluorescence was bleached using a previously published protocol[56] by allowing the tissue to sit in a solution (made of sodium hydroxide and hydrogen peroxide) sandwiched between an LED light panel for 45 mins at RT. The tissue was then washed twice in ddH₂O for 10 mins, followed by a wash in CODEX Hydration Buffer (Akoya Biosciences) twice, 2 mins each. The coverslip was allowed to equilibrate in CODEX staining Buffer (Akoya Biosciences) for 20 mins at RT. Blocking buffer was prepared by adding N, S, J and G blocking solutions to CODEX staining buffer (Akoya Biosciences). Antibodies were then added at respective concentrations to blocking buffer to make a total volume of 100 µl (Supplementary Data 2). The antibody cocktail was added to the coverslip and staining was performed in a sealed humidity chamber at 4 °C overnight. After staining, coverslips were washed twice in hydration buffer for a total of 4 mins followed by fixation in storage buffer (Akoya Biosciences) with 1.6% paraformaldehyde for 10 mins. Coverslips were then washed three times in PBS, followed by a 5 min incubation in methanol on ice for 5 mins followed by another 3 washes in PBS. CODEX fixative solution (Akoya Biosciences) was prepared right before the final fixation step. 20 µl of CODEX fixative reagent was added to 1000 µl PBS. 200 µl of the fixative solution was added to the coverslip for 20 mins followed by 3 washes in PBS. Coverslips were then immediately prepped for imaging or stored in storage buffer at 4 °C until ready for imaging.

**CODEX multicycle setup and Imaging**
The coverslips were mounted onto Akoya's custom-made stage in between coverslip gaskets with the tissue side facing up. The coverslips were cleaned from the bottom to get rid of any salts. The tissue was stained with Hoechst nuclear stain (Thermo Scientific cat # 62249) at a 1:2000 dilution in 1X CODEX buffer. A 96 well plate was used to set up the multicycle experiment with different fluorescent oligonucleotides in each well. Reporter stock solution was prepared containing 1:2000 Hoechst stain and 1:12 dilution of assay reagent in 1X CODEX buffer (Akoya Biosciences). Fluorescent oligonucleotides (Akoya Biosciences) were added to this reporter solution at a final concentration of 1:50 in a total of 250 µl per well. A blank cycle containing no fluorescent probes was performed at the beginning, middle and

end of the imaging cycles to capture autofluorescence (Supplementary Data 2).

Automated image acquisition was performed using Akoya's software driver CODEX Instrument Manager (CIM, version 1.30) and the CODEX platform (Akoya Biosciences). Imaging was performed using a Keyence BZ-X810 microscope, fitted with a Nikon CFI Plan Apo λ 20X/ 0.75 objective. 9-11 z steps were acquired with the pitch set at 1.5 in the BZ-X software. Raw tiff tile images were processed using the CODEX Processor version 1.7.0.6 (version 1.8.3 for qptiff data) by Akoya Biosciences. CODEX Processor subsequently performed drift-compensation, deconvolution and background subtraction for each tile image before finally stitching them into one whole slide image.

**Multiplexed imaging data analysis pipeline**
The multiplexed imaging analysis pipeline included preprocessing the raw images (image stitching, drift compensation, deconvolution, background subtraction and cycle concatenation) using CODEX Processor, cell segmentation using Mesmer[57], cell type identification using CELESTA[25], followed by spatial analysis.

**Cell segmentation**
Protein markers CD45, KRT14, MUC-1, S100A4, ITGAV, CD31, CAV1, PDGFRα/β and FAP were used for whole cell segmentation; DAPI was used as a marker for nuclear segmentation. Mesmer segmentation outputted spatial coordinates of each detected cell as well as the signal intensities for all the protein markers. Downstream analysis was performed using these files.

**Multiplexed imaging data cell type identification**
After cell segmentation, CELESTA R package (https://github.com/plevritis-lab/CELESTA) was used to assign a cell type for each individual cell for each sample. CELESTA takes two inputs, namely (1) segmented data file from Mesmer for each sample with each row as a cell and each column with a protein marker expression and (2) a user supplied cell type signature matrix (Supplementary Data 3) designed to capture the cell type markers in the imaging panel. The cell type signature matrix included common cell type markers for immune cells such as MPO for neutrophils, CD68 for macrophages, CD11c for dendritic cells, and CD3, CD4, CD8, FOXP3 for T cell subtypes. It also included stroma markers such as CD31 for vasculature endothelial cells and FAP and α-SMA for fibroblasts. For tumor cell subtypes, KRT14, KRT6 and TROP2 were used for identifying BST cells, and KRT14, CHI3L1 and TAGLN

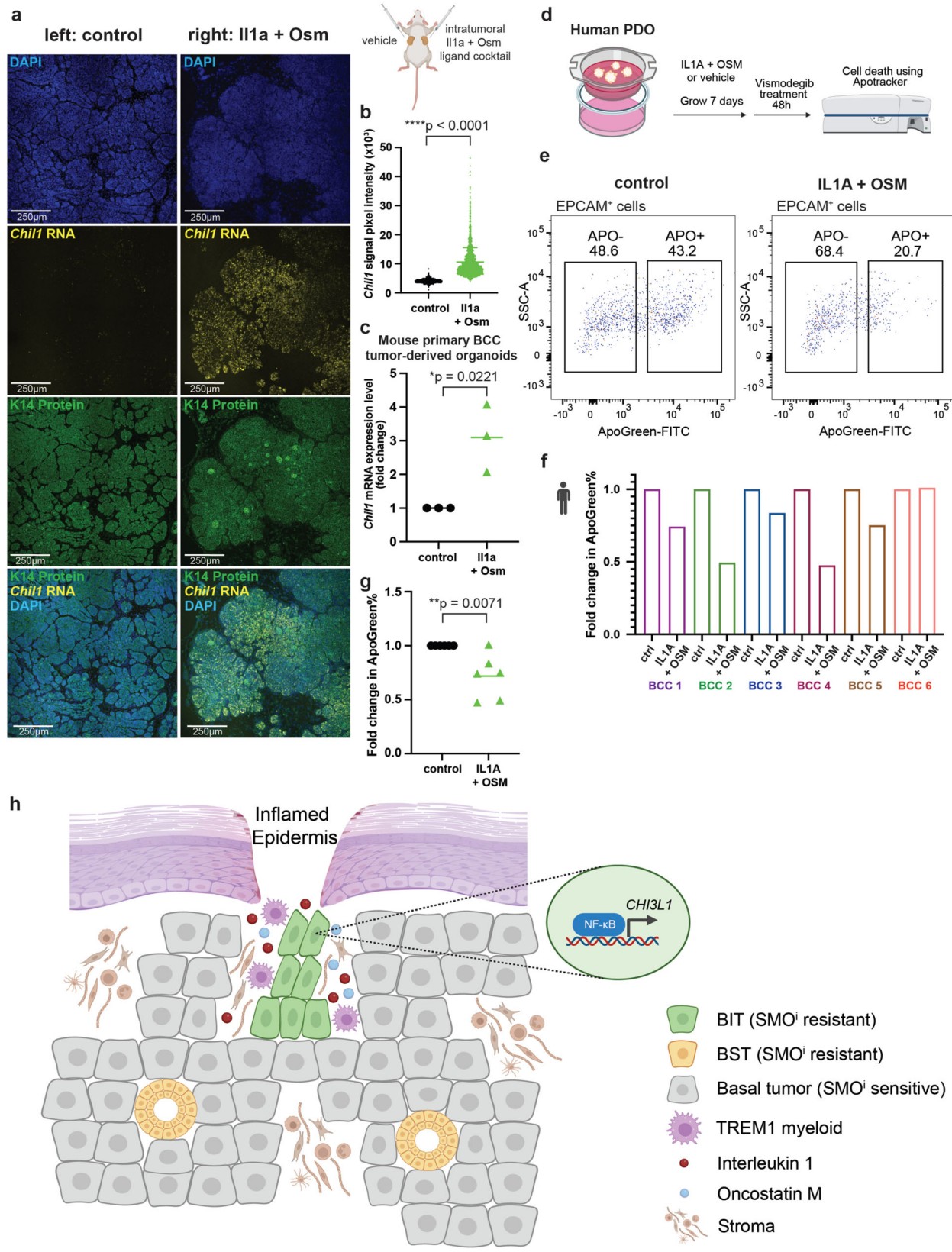

were used for identifying BIT cells. In addition, CELESTA assigned a cell to the "Unknown" category if the cell did not pass the probability threshold to be assigned to a cell type defined in the signature matrix and to account for potential artifacts introduced in the imaging, pre-processing and segmentation steps. The cell type assignments were visualized using the CELESTA R package plotting function.

## Spatial analysis

To identify spatial communities for each sample, we used a similar strategy described previously[26]. Briefly, we first defined a neighborhood for each cell with its 100-nearest-neighbor cells based on Euclidean distances. The number of nearest neighbors was decided based on manual evaluations of the microstructures of the cell types. We

**Fig. 7 | IL1 and OSM combination treatment induces BIT in vivo and ex vivo, leading to resistance to SMO^i therapy. a** Schematic representation of in vivo intratumoral Il1a + Osm ligand cocktail (or vehicle control) injection on allograft secondary BCC tumors in Nod-Scid mice derived from primary mouse BCC tumors from *Ptch1^{+/-};pS3^{f/f};K14Cre-ER;RFP^{f-s-f}* mouse model (*far right*). Intra-tumoral injections were performed at 3-day interval for 18 days. *Left panels* show representative RNAscope images of DAPI, *Chil1* RNA and K14 protein staining of BCC tumors upon vehicle control injection; *Right panels* show representative RNAscope images of DAPI, *Chil1* RNA and K14 protein staining of BCC tumors upon Il1a + Osm ligand cocktail injection. Experiments were performed in *n* = 3 mice (6 tumors). Scale bar = 250 μm. **b** Quantification of *Chil1* RNA expression shown in (**a**) by pixel intensity measurements. Horizontal bars and error bars represent the mean +/- SD. *P*-value calculated using two-sided independent sample T test (****$p < 0.0001$) (*n* = 2180 pixel measurements in control group, *n* = 1799 pixel measurements in Il1a + Osm

group). **c** *Chil1* expression fold change upon Il1 + Osm ligand cocktail treatment compared to PBS control in mouse primary BCC tumor-derived organoids (*n* = 3 biological replicates). *P*-value calculated using two-sided independent sample T test (*$p < 0.05$). **d** Human patient-derived organoid (PDO) treatment strategies with IL1A + OSM or vehicle control. **e** Flow cytometry analysis showing levels of Apo-Green in the EPCAM+ cells from human PDOs. **f** Fold change in ApoGreen percentage between vehicle control and IL1 + OSM treated PDOs (n = 6 independent BCC patient tumors). **g** *P*-value statistic of fold change in ApoGreen percentage calculated using two-sided independent sample T test (**$p < 0.01$) (*n* = 6 independent BCC patient tumors). **h** Summary of overall model. Figure panels 7a, d, f and 7 h are created with BioRender.com released under a Creative Commons Attribution-NonCommercial-NoDerivs 4.0 International license. Source data are provided as a Source Data file.

secondly calculated the proportions for each cell type within each cell neighborhood. We then performed k-means clustering on the cell type proportions across all the neighborhoods for each individual sample. The neighborhoods were over-clustered to 30 clusters for each sample, and then we mapped the neighborhoods back to the tissue images and examined the cell type enrichment for each cluster. For each sample, we merged the 30 clusters into around 9–13 final clusters that we defined as spatial communities. The cell type enrichment score for each community for each sample was defined as the cell type proportion within the cluster over the cell type proportion for all cells in the sample. Each community was assigned a name based on the cell type enrichment scores. To compare the cell type proportions among the three tumor communities (Tumor (non-BIT/BST), BST, and BIT), non-parametric Kruskal-Wallis test for multiple groups was used. The statistical comparisons were performed across all four samples included in the CODEX spatial analysis.

For the distance analysis, for each tumor cell, we calculated its shortest Euclidean distance to other cell types in each sample using the spatial statistical R packages spdep version 1.2-4 and sf version 1.0-7. The distances were converted from segmented pixel values to micrometers using 2.65 pixels per micrometer based on the imaging instrumentation. We compared the distributions of the shortest distances from the three tumor types (tumor (non-BIT/BST), BST, and BIT) to other cell types.

### scRNA-Seq library preparation and sequencing
As per the directions of Chromium Next GEM Single Cell 3′ Reagent Kits v3.1 (Dual Index) user guide (CG000315 Rev C), human or mouse live FACS-sorted cells were washed in PBS containing 0.04% BSA and resuspended in single cell suspension at a concentration of ~1000 cell/μl. We aimed to capture 10,000 cells. Each library was sequenced on an Illumina Novaseq platform to achieve a coverage of ~30,000 reads per cell.

### scRNA-Seq analysis
scRNA-Seq library processing, quality control, and clustering FASTQ files were aligned using 10X Genomic Cell Ranger 3.1.0 and aligned to either an indexed hg38 or mm10 for human or mouse tumors, respectively. Standard workflow using Seurat v3 using R version 4.1.2 were utilized to filter cells. Cells were projected in two-dimensional space using UMAP with the following sample-specific parameters and subset workflow listed below. For human BCC samples, each tumor was initially clustered separately from one another to identify tumor epithelial populations. Based on marker gene expression of epithelial markers (KRT14) and tumor-specific markers (GLI1), we were able to subset tumor epithelial populations for downstream analysis. We were then able to integrate tumor epithelial populations from multiple patients using the Multiple Dataset Integration and Label Transfer (anchoring) standard workflow within Seurat[58]. For the resistant BCC

datasets, we extracted the tumor epithelial populations (as identified through metadata labels) and integrated them together with treatment naïve BCC datasets for expression analysis[19]. Marker genes were found for each individual cluster using the standard Seurat pipeline. For gene scoring, we used the AddModuleScore function built into Seurat.

### scRNA-Seq pseudotime analysis
Pseudotime analysis for tumor epithelial populations was performed with Monocle3[22] to infer pseudotime trajectories based on UMAP coordinates. Tumor state 2 (the least differentiated tumor state) was set as the root node for trajectory analysis.

### scATAC-seq library preparation and sequencing
As per the directions of the Chromium Next GEM Single Cell ATAC Reagent Kits v1.1 user guide (CG000209 Rev D), single-cell suspensions of patient tumors were FACS-sorted for live cells, subsequently washed in PBS containing 0.04% BSA and resuspended in single cell suspension at a concentration of ~1000 cell/μl. Nuclei was isolated according to 10x Genomics demonstrated protocol (CG000169 Rev D). We aimed to capture 10,000 nuclei. Each library was sequenced on an Illumina Novaseq platform to achieve a coverage of ~30,000 reads per cell.

### scATAC-seq library processing, quality control, and clustering
For the scATAC-Seq library processing, 'cellranger-atac count' function (cellranger-atac, v1.1.0) was used for pre-processing of scATAC-Seq data using hg38 human genome. scATAC-Seq analysis was carried out using the R package ArchR (1.0.1). Within ArchR, 'Arrow files' are generated for each sample from the fragment file with the parameters of 1000 unique fragments and a transcription-start-site enrichment value of 4. We used the 'addIterativeLSI' function of ArchR to perform dimensionality reduction on each ArchRproject and applied the Harmony 0.1.0 algorithm to mitigate batch effects for the merged ArchRproject. Samples were visualized using UMAP with the 'addUMAP' function followed by the 'plotEmbedding' function. To subset populations of interest for downstream analysis, the 'subsetArchRproject' function was used. To identify tumor epithelial cells from other cell types, we used the KRT14 GeneScoreMatrix and GLI1 GeneScoreMatrix, which were specific to the tumor epithelial populations. We created pseudo-bulk replicates for each cluster, used MACS2 2.27.1 to identify peaks unique to each cluster, and used 'getMarkers' function to obtain the marker list of each cluster. We then added motif annotation to ArchRproject with the function 'addMotifAnnotations', which identified enriched motifs associated with differential peaks. We applied the 'addImputeWeights' function to impute the weights of marker features and the 'plotEmbedding' function to visualize the marker features. We performed pairwise comparisons of peaks and TFs between clusters using the getMarkerFeatures function.

## Integration of scRNA- and scATAC-seq datasets

Starting with the full dataset, we matched each scATAC-seq cell with its closest corresponding scRNA-seq cell using canonical correlation analysis. Specifically, we used the ArchR function 'addGeneIntegrationMatrix', which employs Seurat's 'FindTransferAnchors' function to integrate datasets[59]. We repeated this integration procedure for subclustered tumor epithelial datasets. For both the full dataset and the subclustered tumor epithelial dataset we observed high correspondence between scRNA- and scATAC-seq-derived cluster labels.

## RNAscope

All RNAscope experiments were performed using the Multiplex Fluorescent v2 system (ACD; 323100) as per manufacture protocols. Briefly, tumors were fixed with 4% paraformaldehyde overnight at 4 °C, paraffin-embedded, and sectioned at 5 µm. We then followed the RNAscope protocol for formalin-fixed, paraffin-embedded tissues as per manufacturer instructions. RNAscope probe Mm-Chil1 - Mus musculus chitinase-like 1 (Chil1) mRNA (ACD; 449621) was used. After completing the RNAscope protocol, tissue sections were immediately blocked, stained with KRT14 antibody (1:500; BioLegend; 906001) followed by staining with the anti-chicken Alexa488 secondary antibody (1:500; Invitrogen; A-11039). Nuclei visualization was with Hoechst 33342 (1:1000; ThermoFisher; 62249). Confocal images were acquired using a Leica SP8 microscope with the Leica LAS X 3.7.6 software. Images were analyzed using FIJI software.

## Visium Sample Preparation, Library Preparation, and Sequencing

Human tumor samples were fixed in 4% PFA overnight, paraffin-embedded, and sectioned at 7 µm for Visium profiling according to 10x Genomics demonstrated protocol (CG000408). Alternatively, fresh tumor tissues were directly embedded in OCT on dry ice and sectioned at 10 µm for Visium profiling according to 10x Genomics demonstrated protocol (CG000240). For FFPE tissue samples, sequencing libraries were prepared using Visium Spatial Gene Expression Reagent Kits for FFPE (CG000407) protocol according to the manufacturer's manual. For fresh frozen tissue samples, sequencing libraries were prepared using Visium Spatial Gene Expression Reagent Kits (CG000239) protocol according to the manufacturer's manual. H&E-stained tissue was imaged using a Keyence microscope in accordance with 10x genomics Visium protocol. The pooled, 3' end libraries were sequenced using an Illumina NextSeq sequencer or an Illumina NovaSeq sequencer.

## Spatial transcriptomic analysis

Space Ranger version 1.3.1 was used for primary data analysis, including demultiplexing, alignment, mapping and UMI counting. Specifically, for alignment and mapping, the GRCh38 reference genome and corresponding annotation were used. Further analysis was performed in using the Seurat R package. The "SCTransform" feature was used to normalize UMI counts. The "FindClusters" function was applied to assign identities based on gene expression. The spatial Seurat objects were scored using the "SpatialFeaturePlot" feature in the Spatial Seurat R package. Gene signatures associated with each annotated cell type in matched scRNA-Seq dataset were used to score spatial Seurat clusters.

## Bulk RNA sequencing and analysis

RNA samples were obtained from adherent cells using RNeasy Plus Mini Kit from Qiagen according to manufacturer's instructions. Library preparation and sequencing were performed as described previously[60]. The RNAseq libraries were constructed by TruSeq Stranded mRNA Library Prep kit (Illumina) and sequenced on an Illumina NovaSeq sequencer. Read alignment was performed using STAR[61] with mm10 as reference genome. Raw counts and FPKM values were

called using the HOMER function "analyzeRepeats.pl". The DESeq2 R package was used to generate differential gene expression. Gene lists were submitted to https://geneontology.org/ for Gene Ontology enrichment analysis. Gene signature enrichment in each sample was calculated based on ROAST gene set testing[62] and logFCs visualized using a barcode enrichment plot with limma package (Bioconductor). The read counts of differential gene lists were normalized using quantile normalization and heatmaps were generated by clustering genes using the "heatmap.2" function from gplots R package.

## ATAC sequencing library preparation, sequencing and analysis

ATAC-seq was performed as described previously[63,64]. In brief, 100,000 ASZ-001 cells were plated per replicate in a 48-well plate, then serum starved for 24 h prior to transposition. Cells were washed with cold PBS and lysed using 0.1% NP40 in resuspension buffer for 10 mins at 25 °C. Cells were washed again, then 100 µl of transposition mixture was added to each reaction and incubated at 37 °C for 30 mins, shaking at 1000 RPM using a thermo-shaker. DNA was purified using QIAGEN MinElute PCR Purification Kit and then amplified for 8–15 cycles to produce libraries for sequencing. The libraries were initially sequenced on an Illumina MiSeq sequencer and analyzed using a custom script to determine the Transcriptional Start Site (TSS) enrichment score by calculating the ratio of signal over background at TSS over a 2-kb window. Only libraries that had the highest score above the threshold (> 5) were chosen for deeper sequencing. Two independent, biological replicates were sequenced on an Illumina NovaSeq sequencer. Paired-end reads were trimmed for Illumina adapter sequences using Trim Galore and mapped to mm10 using Bowtie2 2.3.4.1[65] with parameters -p 4–very-sensitive. Duplicate and mitochondrial reads were discarded with Samtools v0.1.18. MACS2[66] with the parameters −nomodel −extsize 73 −shift −37 and FDR threshold 0.01 was used for peakcalling, and background removal was carried out via submitting replicates to irreproducible discovery rate (IDR) filtering[67]. Overlapping peaks from all samples were merged into a unique peak list, and raw read counts mapped to each peak [using bedtools multicov (Quinlan laboratory, University of Utah, Salt Lake City, UT)] for each individual sample were quantified. Differentially accessible peaks from the merged union peak list were identified with DESeq2 R package. Cutoffs were set at $\log_2$ fold change >1 or <−1 and P-value < 0.00001. Read pileups at genomic loci were imaged using Integrative Genomics Viewer (Broad Institute).

## CUT&RUN library preparation, sequencing and analysis

CUT&RUN was performed with the CUT&RUN Assay Kit (86652, CST) according to manufacturer's protocol. Briefly, cells were harvested, washed, and bound to activated Concanavalin A-coated magnetic beads and permeabilized. The bead-cell complex was incubated overnight with either an anti-NFKB1 antibody (13586, CST) or an IgG antibody control at 4 °C. Cells were then washed with digitonin buffer, resuspended in 50 µl of pAG/Mnase and incubated for 2 h at 4 °C. Enriched DNA fragments were purified using QIAGEN MinElute PCR Purification Kit. CUT&RUN libraries were subsequently prepared using NEBNext Ultra II DNA Library Prep Kit for Illumina according to manufacturer's protocol. Two independent, biological replicates were sequenced on an Illumina NovaSeq sequencer. Paired-end reads were trimmed for Illumina adapter sequences using Trim Galore and mapped to mm10 using Bowtie2 2.3.4.1[65] with parameters -p 4–very-sensitive. Duplicate reads were discarded with Samtools v0.1.18. MACS2[66] with FDR threshold 0.01 was used for peakcalling and background removal was carried out via submitting replicates to irreproducible discovery rate (IDR) filtering[67]. Overlapping peaks from all samples were merged into a unique peak list, and raw read counts mapped to each peak [using bedtools multicov (Quinlan laboratory, University of Utah, Salt Lake City, UT)] for each individual sample were quantified.

## Motif analysis of peaks from ATAC-seq and CUT&RUN

Motif analysis on peak regions was performed using HOMER function (http://homer.ucsd.edu/homer/motif/) "findMotifsGenome.pl" with default parameters to calculate the occurrence of a TF motif in peak regions compared to that in background regions. We used -$\log_{10}$ (*P*-value) to rank the enrichment level of TF motifs.

## Flow cytometry/fluorescence-activated cell sorting (FACS)

For FACS or flow experiments, single-cell suspensions of cells were resuspended in 2% FBS FACS buffer and then stained with appropriate antibodies at a 1:100 dilution for 60 mins at 4 °C in the dark. Antibodies used were PE/Cy7 anti-human CD326 (Biolegend; 324222) and BV711 anti-human CD90 (Biolegend; 328139). Cells were subsequently washed twice using FACS buffer and strained through a 40 μm filter. Before sorting or flow, SytoxBlue (Thermo Fisher; S34857) was used for a viability dye at a 1:1000 dilution. FACS experiments were run on FACSAria II, and flow experiments were run on Symphony 2 instruments. Both instruments used the BD FACSDiva 8.0.1 software for data collection. For scRNA-Seq experiments, cells were only stained with SytoxBlue, and live cells were sorted for downstream processing and experimentation. For Apotracker™ experiments, we used the Apotracker™ Green reagent as per manufacturer instructions (BioLegend; 427402). FlowJo version 10.6 was used for data analysis.

## In vitro ligand screening assay

ASZ-001 cells were treated with the following concentrations of recombinant ligands in serum-free medium for 24 h or 48 h: 25 ng/ml of Il1a, 25 ng/ml of Il1b, 20 ng/ml of Ifn-γ, 25 ng/ml of Il6, 50 ng/ml of Osm, 50 ng/ml of Tweak, 5 ng/ml of TGF-β, 25 ng/ml of Il1a + 50 ng/ml of Osm, 25 ng/ml of Il1b + 50 ng/ml of Osm, 25 ng/ml of Il1a + 25 ng/ml of Il6, 25 ng/ml of Il1b + 25 ng/ml of Il6, or PBS vehicle control. Following ligand treatments, cells were harvested for subsequent analysis using qRT-PCR and bulk RNA-Seq analysis. To determine the reversibility of ligand effects, cells were treated with 25 ng/ml of Il1a and 50 ng/ml of Osm for 24 h before removing the ligands for 48 h or removing the ligands and treating the cells with 100 ng/ml of Il1 receptor antagonist for 48 h. Following cytokine withdrawal/antagonism, cells were harvested for subsequent analysis using qRT-PCR.

## In vitro drug treatments

In each treatment arm, ASZ-001 cells were pre-treated with the following concentrations of drugs respectively in serum-free medium for 6 h before adding 25 ng/ml of Il1a and 50 ng/ml of Osm to the cells for 24 h: 20 μM of NFKB inhibitor JSH23, 1 μM of NFKB inhibitor withaferin A (WFA), 1 μM of JAK/STAT3 inhibitor ruxolitinib or DMSO control. Following drug and ligand treatments, cells were harvested for subsequent analysis using qRT-PCR.

## In vivo intra-tumoral injection

Intra-tumoral injection experiments were performed on allograft secondary BCC tumors in adult female Nod-Scid mice (7–8 weeks of age) derived from primary mouse BCC tumors from X-ray irradiated *Ptch1*[+/−]; *p53*[f/f];*K14Cre-ER;RFP*[f-s-f] mouse model. 30 μl of PBS vehicle was injected into the left tumor and 30 μl of Il1a (500 ng) + Osm (500 ng) ligand cocktail was injected into the right tumor of the same mouse at 3-day interval for 18 days. Experiments were performed in *n* = 3 mice (6 tumors).

## Patient-derived organoid (PDO) model

Freshly resected tumors were obtained from 6 de-identified patients undergoing dermatological surgery. The PDO model was based on previous established protocols for culturing pieces of tumor and culturing them long term[68]. Briefly, Millicell-CM inserts (MilliporeSigma; PICM03050) were prepared with 1 ml of an 8:1:1 ratio of Cellmatrix

Type 1-A (Fujifilm; 637-00653), 10× concentrated culture media HAM's F-12 (catalog), and reconstitution buffer. Minced human tumor specimens were re-suspended in 1 ml of Cultrex Reduced Growth Factor Basement Membrane Extract, Type R1 (Biotechne; 3433-010-R1) and then placed on the solidified 8:1:1 collagen matrix. 1 ml of PDO media in Advanced DMEM/F-12 (Gibco; 12634010), supplemented with: 50% WNT3A, RSPO1, and Noggin conditioned media (ATCC; CRL-3276), HEPES, (1 mM; Gibco; 15630080), GlutaMAX (1x; Gibco; 35050061), N-Acetylcysteine-1 (1 mM; Sigma Aldrich; A9165-5G), penicillin/streptomycin (100 U/L; Gibco; 15140122), nicotinamide (10 mM; MilliporeSigma; N3376-100G), B-27 Supplement (50×) minus vitamin A (1x; ThermoFisher; 12587010), A83-01 (0.5 μM; Tocris; 2939), Gastrin I human (10 nM; MilliporeSigma; G9020-250UG), SB202190 (10 μM; MilliporeSigma; S7067-5MG), Recombinant human IL-2 (1000 units/ml; Peprotech; 200-02), Human EGF (50 ng/ml; ThermoFisher, PHG0311L), Recombinant Human M-CSF (20 ng/ml; Peprotech; 300-25). L-WRN cells (ATCC; CRL-3276) used to generate conditioned media cells were monitored by their morphology. No mycoplasma was detected through regular testing.

For ligand treatments, at the time of plating, a combination of 25 ng/ml IL-1 and 50 ng/ml OSM (or PBS vehicle control) were added to the organoid media. For acute drug treatments, organoids were first plated for 7 days, and Vismodegib (1 μM; Selleckchem; S1082) was added for 48 h before collecting samples for analysis. Media were replaced every 3 days. At experimental endpoint, the tissues from each well were further dissociated into single cells using 5 ml of 0.25% collagenase solution (Gibco; 17-100-017) in DKFSM media (Gibco; 10744-019) incubated at 37 °C with rotation for 30 min to first dissociate the matrix that the PDOs are suspended in. PDOs were pelleted, the 0.25% collagenase solution removed and then replaced with 5 ml of fresh 0.5% collagenase solution incubated at 37 °C with rotation for 1 h. At the end of the 1 hour incubation, 0.5 ml of 0.25% Trypsin (Gibco; 25200056) was then added to further dissociate for an addition 10 mins before antibody staining and flow analysis. Apoptosis of the epithelial cells was assessed with Apotracker Green (Biolegend; 427402). For qPCR analysis, tissues were homogenized using a tissue homogenizer (Thermo Fisher Scientific, Waltham, MA) before RNA extraction as previously described[20]. RNA extracts were used to carry out qRT-PCR with TaqMan probes (Thermo Fisher Scientific, Waltham, MA).

## Survival analysis using human TCGA cancer data

To examine how expression levels of Tumor state 3 (TS3) gene signature in various tumor types correlate with patient overall survival, we utilized the GEPIA web interface (http://gepia.cancer-pku.cn/).

## Quantification and statistical analysis

In general, data represent results from three or more independent biological samples, unless otherwise described. Deep sequencing data are from two biological samples per condition. Quantification of fluorescence signals from RNAscope images was performed on *n* ≥ 40 cells per condition. Quantification of pixel intensity was performed using FIJI software (NIH; Bethesda, Maryland)[69]. Bar and line graph results reflect the mean with standard deviation (SD). Statistical comparisons were performed using unpaired two-sided Student's t-test unless otherwise specified. The software used for statistical analysis is GraphPad Prism (La Jolla, CA), with the annotations: ns, non-significant, *$p < 0.05$, **$p < 0.01$, ***$p < 0.001$, ****$p < 0.0001$. Details of statistical methods for specific analysis are described in corresponding methods and figure legends.

## GRAPHICS

The diagrams in Figs. 1a, 7a, d, f, and h were created by the authors using BioRender.com.

## Reporting summary

Further information on research design is available in the Nature Portfolio Reporting Summary linked to this article.

## Data availability

The human-specific sequencing data generated from this study (scRNA-Seq, scATAC-Seq and Visium spatial transcriptomics data) have been deposited in the dbGaP database under the accession code phs003437.v1.p1. As these are raw sequencing datasets for human patients, dbGaP ensures that only authorized researchers such as tenure-track professors or senior scientists, working on similar studies with appropriate approvals can access and analyze it. Principal investigators interested in the raw datasets should submit a request for access to dbGAP study. Investigator's institutional Signing Official and NIH Data Access Committee need to approve the request to obtain access, and the process might take 2–4 weeks. Once the request is approved, the investigator will be allowed to access the dataset for one year, with the option to renew. The mouse-specific sequencing data generated from this study have been deposited in NCBI Gene Expression Omnibus database under the accession code GSE248314. Previous human BCC scRNA-Seq data and scATAC-Seq data used in this study[18] are available in dbGaP database under the accession code phs003103.v1.p1. The resistant human BCC scRNA-Seq publicly available data used in this study[19] are available in the NCBI Gene Expression Omnibus under the accession code GSE123814. Additional human BCC scRNA-Seq datasets used in this study[20,21] are available in Gene Expression Omnibus database under accession codes GSE181907 and GSE141526. The inflammatory skin diseases scRNA-Seq publicly available data used in this study[29] are available in ArrayExpress under the accession code E-MTAB-8142. The alopecia areata scRNA-Seq publicly available data used in this study[30] are available in the NCBI Gene Expression Omnibus under the accession code GSE212450. The colorectal cancer scRNA-Seq publicly available data used in this study[32] are available in the NCBI Gene Expression Omnibus under the accession code GSE132465. The pan-cancer scRNA-Seq publicly available data used in this study[31] are available in the NCBI Gene Expression Omnibus under the accession code GSE154763. The diabetic foot ulcer scRNA-Seq publicly available data used in this study[33] are available in the NCBI Gene Expression Omnibus under the accession code GSE165816. The human metastatic urothelial cancer scRNA-Seq publicly available data used in this study[36] are available in the NCBI Gene Expression Omnibus under the accession code GSE145281. The human biliary tract cancer CITE-Seq publicly available data used in this study[37] are available in the NCBI Gene Expression Omnibus under the accession code GSE210067. Raw CODEX images are available upon request to corresponding authors due to the huge file sizes of raw data files. The remaining data are available within the Article, Supplementary Information and Source Data file. Source data are provided with this paper.

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

## Acknowledgements

We thank Pauline Chu, BS, HT (ASCP) from Stanford Human Histology Research Core, Wesley Alejandro and Anum Khan from Stanford University Cell Sciences Imaging Core Facility (RRID: SCR_017787). The following NIH Shared Instrument Grants were used to fund equipment used in the Stanford Shared FACS Facility (RRID: SCR_017788): NIH S10RR025518-01 and NIH S10RR027431-01. The following NIH Shared Instrument Grant was used to fund equipment used in the Stanford University Cell Sciences Imaging Core Facility (RRID: SCR_017787): NIH 1S10OD010580. Visium processing and sequencing were carried out at Stanford Genomics (supported by NIH S10OD025212 and 1S10OD021763) and UC Irvine Genomics, Research, & Technology Hub (supported by P30CA-062203). The work is funded by the following grants: N.Y.L. is supported by the Singapore A∗STAR graduate fellowship National Science Scholarship; D.H. is supported by NIH 5T32AR7422-37 and NIH 1F32CA254434; A.R.J is supported by NIH T32ARO07422-40; W.Z and S.K.P are both supported by grants U54-CA209971 and U54-CA274511; A.E.O. is supported by NIH 1R01ARO4786 and NIH 2R37ARO54780.

## Author contributions

N.Y.L. and A.E.O. conceived and designed experiments. N.Y.L., D.H., A.R.J., and C.P. performed experiments. N.Y.L., W.Z., D.H., and S.G. performed data analysis. N.Y.L. wrote the original draft of the manuscript. All authors contributed to the interpretation of results and revisions to the manuscript. Funding for this project was acquired by S.K.P. and A.E.O. The project was under the supervision of S.K.P. and A.E.O.

## Competing interests

The authors declare no competing interests.
