## [Peer Review File · Nature Communications]

Basal-to-inflammatory transition and tumor resistance via crosstalk with a pro-inflammatory stromal nicheReviewers' Comments:

Reviewer #1:

Remarks to the Author:

In this manuscript, Li et al. profiled human skin basal cell carcinoma (BCC) samples (n = 4 total, consisting of n = 2 naïve and n = 2 smoothed inhibitor treated) with scRNA-seq, scATAC-seq, multiplexed imaging (CODEX, 41-protein panel), and spatial transcriptomics (10X Visium). Their analysis identified a novel subpopulation of tumor cells (or a tumor state) that they describe as basal to inflammatory transition (BIT), which is enriched in SMOi treated samples. BIT is marked by CHI3L1, TAGLN, ITGAV, and VCAM1 and seems to downregulate hedgehog pathway signaling (lower GLI expression). Integration of scATAC-seq data highlighted enrichment in NFKB family TF motifs within open chromatin of BIT cells. CODEX demonstrated that BIT cells are enriched at the superficial aspect of BCC tumors, but distinct from basal to squamous transition (BST) cells, a drug-resistant state that authors identified in earlier work. BIT cells are adjacent to tumor stroma consisting of myeloid cells such as neutrophils and macrophages. Conversely, BST cells are located more toward the tumor core and away from stroma. Authors then performed cell-cell communication analyses using scRNA-seq data and CellChat. This highlighted TREM1+ myeloid cell communication to BIT cells through IL1 and OSM. Visium data appeared consistent with CODEX findings, including BIT high cells being more superficially located than BST cells, with BIT cells in the vicinity of TREM1 myeloid cells. Finally, authors experimentally validated some of these findings using mouse BCC cell lines, which demonstrated that RNA-seq of combined IL1A+OSM treated BCC cells upregulated genes from the BIT signature, with ATAC-seq data highlighting enrichment of NFKB1 motifs within open chromatin after IL1A+OSM treatment. CUT&RUN with anti-NFKB1 supported direct binding of NFKB1 to the CHIL1 enhancer. Authors additionally performed in vivo experiments with intratumor injection of IL1A+OSM, which upregulated CHIL1. Human patient-derived organoids were able to resist SMOi-induced apoptosis in the presence of IL1+OSM. Overall, these studies identified a novel drug-resistant tumor state (BIT) distinct from the already established BST SMOi-resistant state, which seems at least partially regulated by OSM and IL1A that stem from TREM1+ myeloid cells. This study nicely combined several different single-cell and spatial profiling approaches of human samples to generate hypotheses regarding cell-cell communication in the TME, which were validated experimentally in both mouse and human models of BCC. The study could benefit from additional rigor of some analysis and experiments to more convincingly support their claims. I detail some concerns below.

Major comments:

1. I had trouble following how many samples were profiled for each technique. Based on Fig. 1a, 5 tumor samples were profiled for scRNA-seq and/or scATAC-seq, but at most 4 samples seem to be shown at a time. It is unclear if only some samples were profiled with both scRNA-seq and scATAC-seq or if the scATAC-seq were distinct samples. CODEX data is shown for 2 samples (1 in main figure, 1 in supplement), yet it seems 4 were profiled and analyzed. Same goes for the Visium data. This raises a few questions:
 - a. How universal are these findings (BIT in consistent areas across all samples) and how do they vary across patients in spatial data? Please show data from all samples to provide clarity.
 - b. Nearest distance analyses and density plots also seem to be only done a subset of samples (Fig.

3h, S3d).

c. Authors identified 17 cell types in CODEX (Fig. S3a), but most of the time, only a subset of them are shown in each CODEX sample (e.g., from Fig. 3d). Does each sample only contain a subset of identified cells? Are some cell types not present in certain samples?

2. The BIT signature genes shown in Fig. 1f contain some genes that resemble epithelial to mesenchymal transition (EMT) or partial EMT, such as LAMB3, LAMA3, LAMC2, TGFBI among others. Yet, authors were quick to call this BIT based on implication of active NKFB signaling. Was GO analysis performed on the signature to support only an “inflammatory” state? Or are additional states present, i.e., some mixing of states happening? In Fig. 4b-c (CellChat analyses) TGFb (inducer of EMT) is prominently active, but authors seemed to ignore this.

3. Fig. 4h, can authors employ statistical testing on the BIT low vs. high tumors to show a significant difference in TREM1 myeloid cells?

4. In IL1A+OSM recombinant protein experiments, how much of the BIT signature is upregulated by IL1A+OSM? It looks partial at most, as some of the genes I mentioned above that were found in human data (LAMB3, TGFBI, etc.) are not shown in Fig. 6c.

5. Fig. 6e, GSEA plots look very “off.” Authors should double check that GSEA was done correctly.

6. There is a discrepancy between human and mouse data, as CHI3L1 appears to be the human BIT marker, while CHIL1 is shown for any mouse BCC experiments. Authors should address this difference.

7. I do not see any IgG antibody control CUT&RUN data to demonstrate specificity of the NKFB1 antibody (Fig. 6h).

8. Authors’ experiments show that IL1A+OSM are at least partially sufficient to induce BIT and NKFB1 genome binding and activity inferred through motif analyses, but experiments do not show whether NKFB1 is required for BIT. I understand that the models are limited in that they do not spontaneously recapitulate BIT, but it would be more convincing to show a functional role of NKFB1 through KO or KD experiments.

9. While I note that raw sequencing data should be deposited in protected databases such as dbGAP, processed data such as counts matrices (or cell ranger outputs) for scRNA-seq/scATAC-seq should be included in the GEO submission, which is currently missing. Similarly processed spatial data should be deposited as well (at least for 10X Visium). It would be nice if authors made CODEX data available somehow as well.

Minor comments:

1. P. 13, variations should be variance if referring to PCA

2. Fig. 7f should be shown as paired samples and statistics applied.

Reviewer #2:

Remarks to the Author:

Summary: In this manuscript the authors have applied single-cell RNA-seq and ATAC-seq, the visium platform to perform spatial transcriptomics, and multiplexed imaging on human basal cell carcinoma samples, to define a new tumor microenvironment which the authors have termed BIT (basal to inflammatory transition). The authors use their single-cell assays to define a BIT signature

and find that NFKB is involved in BIT. The authors suggest that BIT is initiated in a special immune-cell infiltrated inflammation site within tumors, and may contribute to drug resistance to the smoothed inhibitor.

Strong points: The multipronged approach taken by the authors to characterize heterogeneity in basal cell carcinoma is commendable. The authors have used elegant tools to analyze and identify a new microenvironment. This is potentially interesting and important in finding new therapeutic avenues. The comments below can strengthen the conclusions stated in the manuscript.

Major concerns:

1. The authors state throughout the manuscript that BIT emerges within a specialized inflammatory environment. However, how BIT form in tumors has not been thoroughly characterized in the current manuscript.
2. With the single-cell RNAseq data, the authors have defined a tumor state 3, which is claimed as being enriched by drug treatment. Firstly, this is only supported by Fig 1c. Secondly, the conclusion is based on the comparison of two treated tumor samples and two naïve tumor samples. It is understandable that applying all the single-cell approaches to a large number of samples is not feasible. The authors could change the title of Figure 1 (which currently states 'Integrated single cell analysis of human BCC tumors identifies a novel tumor state (TS3) enriched by drug treatment'), since majority of the figure describes analysis from tumor samples that are not classified as treated or untreated.
3. The authors have used IL1 and OSM treatment in mouse BCC cells and performed RNA-seq, ATAC-seq, and CUT&RUN for NFKB1. The analyses of these datasets can be elaborated. For example, the authors can include heatmaps to show genomewide changes in chromatin accessibility and NFKB1 binding. The authors can also define clusters of genomic sites based on genes forming BIT v/s BST signature to strengthen their claim that BIT is distinct from BST.
4. Figure 7 convincingly shows that IL1 and OSM treatment reduces the sensitivity of SOMi treatment. However, the authors have not characterized how the expression of BIT markers changes upon SOMi treatment in IL1/OSM + and – organoids. The authors could perform RNAseq, ATAC-seq and NFKB1 CUT&RUN in the organoid system to dissect how IL1+OSM leads to resistance.

Minor concerns:

1. In line: Notably, among BCC tumor states, tumor state 3 was enriched in SMOi-treated BCC tumors relative to naïve BCC tumors (Fig. 1c, Supplementary figure 1d). Supp 1d does not support this statement.
2. Some of the colours used to highlight the different cell-populations in the CODEX image (Fig 3) are very similar. Eg. BIT and epidermis in 3e. If possible the authors should use more contrasting colours.

3. In the discussion, the authors state 'and we now extend our initial observations through quantitative spatial expression analysis to confirm an optimal distance of 2-3 cell diameters (150-250 μm) from the microenvironment prior to the onset'. Where is the measurement coming from? It was not described in the results section.

Reviewer #3:

Remarks to the Author:

In this article, Li et al. characterize a novel inflammatory signature enriched in therapy-resistant basal cell carcinoma (BCC). Since BCC is the most common human cancer, the research presented in this study is highly relevant and significant. In addition, due to the lack of effective therapeutic options for advanced BCC, finding new pathways to target tumor resistance is imperative. Despite BCC being one of the most mutated cancers, it is usually not highly immunogenic, and immunotherapy is only effective in a subset of patients. Characterization of the immunomodulatory events happening in BCC in human skin will help us better understand how these tumors develop and progress beyond the regulation of Hedgehog signaling.

The article focuses on a novel basal-to-inflammatory transition in cancer cells present in BCC and enriched in therapy-resistant tumors. The authors go into detail characterizing this inflammatory transition and the localization of tumor cells with this signature in the tissue context, comparing it to a previous resistance mechanism identified by the same group (called basal to squamous transition). They identify the causal effect for the signature as being cytokines secreted from myeloid cells that surround the tumor and cause activation of NF κ B transcription in cancer cells. Then, they recapitulate this inflammatory signature by treating mouse and human BCC cells with IL1 α and OSM cytokines and show that the newly acquired phenotype leads to reduced apoptosis following SMO inhibitor treatment, the first line of therapy for advanced BCC patients.

The study is novel since it defines a previously unknown inflammatory signature unrelated to other skin conditions like eczema and psoriasis that could be involved in therapeutic resistance. The detailed analysis of human patient samples also makes the study highly relevant. The article presents complex data analysis in a way that is easy to understand and follow, and all methods and figures are explained in detail.

Some minor additional data and experiments might make the manuscript and the point stronger.

Comments:

- Although it is presented that the BIT signature can lead to therapy resistance by blocking apoptosis following SMO inhibitor treatment, no evidence is presented regarding the mechanisms by which this resistance might happen. Does cytokine treatment influence cell proliferation or only apoptosis resistance? Is there a change in proliferation or apoptosis in the tumors shown in Fig. 7a?
- One critical aspect is whether the BIT phenotype depends on the continuous activation of NF κ B signaling. Is the phenotype reversible or stable? This key detail would determine if BIT can be therapeutically targeted. Could combining cytokine or NF κ B inhibitors with SMO inhibitors revert the resistance phenotype caused by IL1 α and OSM? Showing that the phenotype is reversible by drugs and/or cytokine withdrawal could be easily done in the mouse BCC cell line.
- In line with the previous point, it is unclear if the increased number of BIT cells in therapy resistant BCC is due to an expansion of the BCC BIT cell population or an increased recruitment of TREM1

myeloid cells after treatment. So, does therapy expand previously existing BIT BCC cells, or does it lead to TREM1 myeloid cell recruitment that preserves and induces additional BIT BCC cells? It might be a difficult question to answer, but is it possible to analyze the proportion of TREM1-positive myeloid cells in naïve vs therapy-resistant BCC? This information is presented in suppl. Fig 5i and j for other cancer types but not for BCC.

- What would be a possible mechanism by which activation of NF κ B leads to a loss of Hedgehog signaling? Fig. 5e shows a reduction in Hedgehog signature in cells, but it is not clear what precise components of the pathway are altered. Is there a reduction in the expression of Hedgehog components or a loss of transcriptional activity of Gli? This could be easily analyzed by looking at the expression of Hedgehog signaling components in the RNAseq data and the Gli binding sites in the ATAC-seq.

- The BST cells are located close to the skin surface. Could there be a correlation between the loss of epithelial barrier function and BIT acquisition? For example, authors could analyze if this signature is also present in wound healing datasets. Is it known whether vismodegib treatment leads to skin barrier dysfunction that could increase the recruitment of myeloid cells?

Reviewer #4:

Remarks to the Author:

Review for Li et al. Basal-to-inflammatory transition and tumor resistance via crosstalk with a pro-inflammatory stromal niche

Li et al. profile several treatment naïve patient samples using scRNAseq, scATAC, CODEX and Visium. They identify a tumor epithelial cell state, basal to inflammatory transition (BIT). They show that BIT is spatially localized in a distinct pro-inflammatory niche near TREM1 myeloid cells. They further demonstrate in both cell lines and organoids that IL1 and OSM secreted by TREM1 myeloid cells activate NF- κ B within the BIT tumor epithelium, leading to transcriptional upregulation of BIT markers. Finally, they perform preliminary experiments in vivo to demonstrate that the BIT population confers increased resistance to SMO therapy.

Overall, the paper is interesting and coherent, proposing novel tumor-stroma interactions in basal cell carcinoma with potential therapeutic implications. The authors acquired diverse data on the human samples and did in vitro and in vivo work to provide mechanistic context for their results. I have some concerns, primarily regarding the analysis of the human data. I believe that the paper will be of interest to the readership of Nature Communications after the authors address the following the comments:

1. The paper is very vague on the number of samples that they had, which of these underwent which analytical assay and which of these is presented in each figure/panel. I would like to see a supplementary figure/table summarizing this information. It should also be displayed more clearly in figure 1.
2. The major limitation of the human work is that for the most part it is based on 2-4 patients, depending on the assay. This is really a very low number to say anything about a subject as complex as the tumor microenvironment. While I understand that it is difficult to obtain many samples, and

profile them with a large variety of technologies, perhaps the authors could suffice with a lower-plex IHC or IF panel to examine the existence and location of BIT in additional samples.

3. The authors should better acknowledge the limitation of the small sample size in their work and tone down many of their statements. Examples include (but are not limited to):

a. "Our analysis corroborated previous work demonstrating that BST tumor epithelium occurs centrally and compactly within tumor nodules and does not interface directly with stromal elements"

b. "BIT tumor epithelium arises within a pro-inflammatory myeloid dominant community, spatially and cellularly distinct from the stroma-deplete BST community".

4. Legends throughout the paper are very concise and not informative. Many elements in the figure are not explained. Please provide more information. Examples include (but are not limited to):

a. Fig. 1H: What do the dashed lines represent? Are these confidence intervals? It's not stated in the plot or legend.

b. Fig3h, what does the vertical line mean

c. Fig6c, what does $\log(\text{baseMean})$ mean?

d. Fig4k, why is the y-axis range is so low? What is shown there?

5. Fig. 1 The authors define TS3 as a state that is associated with poor prognosis, but they do not provide any information regarding what this state is, other than saying that it involves expression of CHI3L1, TAGLN, ITGAV and VCAM1. What is TS3? What are the other states that the authors identify in the tumors?

6. The authors saw a rise in NF-kB signaling in the 2 treated patients, suggesting a correlation with the rise of TS3. Since this finding is based on only two patients, can the authors check the other datasets with TS3 to see if they also have more NF-kB signaling?

7. Fig. 2A,B: The authors perform monocle pseudotime analysis, but is there any rationale to doing this? Pseudotime applies when examining a developmental process. Do the authors think that the compendium of states observed in the tumor are all a process of development? I would naïvely assume that some of the differences in states have to do with the different microenvironments of the tumor. What is the meaning of a starting state and end states in the context of a tumor? Why are these not just cooccurring states?

8. Fig. 2D: It is unclear from the methods how the scATAC and scRNAseq data were merged. The UMAP in 2D, showing the merged data looks strikingly similar to the UMAP in 2C which was constructed only based on the scATAC data. How come?

9. Throughout figure 2 the authors do not provide meaningful information regarding the cell states that were derived from the scATAC or scATAC+scRNAseq. They simply refer to them as C1-C5 or "Tumor state1 – Tumor state5". This is not very informative. What do I as a reader learn from this analysis? The authors should do the minimal level of analysis, which is to annotate the clusters that they find and relate it back to the literature.

10. Why is the UMAP in 4A different the one in figure 1E? The legend suggests that only a subset of the tumor cells were taken for the cellChat analysis. Why not all of them? Similarly, why are the values in the graphs in 4B and Supp. 4C different? For example, BIT in 4b is $\pm(1.5,2)$ whereas BIT in Supp. 4c is $\pm(2.1,2.2)$.

11. In figure 4(h,i) the authors claim that the presence of the BIT state in the tumor cells correlates with TREM1 Myeloid cells. To show that, they classify 4 tumors into two high BIT (4.6% and 5.8%) and two low BIT(1.6% and 1.8%). They show that in the BIT high group the TREM1+ are $\pm 60\%$ of the

myeloid cells, whereas in the BIT low they are $\pm 25\%$. I find this analysis problematic for several reasons:

- a. These are only 4 patients, yet the authors group them in 4h to the two groups and just show a single bar graph for each one. They should show a bar graph for each one of the patients.
- b. Similarly, for the UMAP of the Myeloid cells, the authors should show it colored by patients.
- c. Are any of these 4 patients the same as the two patients that were described in figures 1-3 4a-f? It is not clear enough from the text. If it is not the same patients, then there is a mistake in Figure 1A, and it should say that the paper had 8 patients (2 naïve, 2 drug resistant, 4 naïve for validation). If it is the same patients, then the analysis in 4k is misleading. If these are the same patients, then by definition we would expect to see e.g. IL1A in TREM1 Myeloids, because this is how the CellChat analysis found it. If this is an independent validation then it should be conducted on different patients.
- d. The changes in TREM1 myeloid cells are big (25%-60%), yet the changes in the BIT fraction are minuscule (2%-5%). How does this fit?

12. The authors suggest IL1 and OSM secretion from TREM1 Myeloid cells as important signals influencing BIT. They do not relate this at all to the literature. IL1 is known to induce inflammation in cancer in general and in skin cancer in particular. It was already suggested as a signaling pathway that can contribute to resistance mechanisms (e.g. DOI 10.1016/j.cyto.2022.155850 and 10.18632/oncotarget.12590). The same holds for IL6, which OSM is part of the IL6 family (e.g. DOIs 10.1158/0008-5472.CAN-14-2061). There is also work showing that blocking IL1 and IL6 increase anti-tumor activity and that NFkB is involved in this signaling pathway. The authors should put their results in the context of the existing literature.

13. Fig. 5A, same comment as above. Since these are only 4 patients, there is no reason to group them. The authors should kindly show the expression in each patient.

14. Fig. 5: According to the authors, this analysis is “validating the spatial association between BIT tumor epithelium and the specialized TREM1 myeloid cell-associated inflammatory environment” (page 10). However, this Visium analysis is performed on the same patient on which the single-cell and CODEX analyses were performed. As such, it doesn’t provide a general validation for the association of these cell types across patients. The authors should tone down their statement to reflect the fact that this is an additional way to look at the same tumor.

15. Fig. 5G: The authors compare their TREM1 signature to two datasets of healthy skin, eczema, psoriatic and alopecia areata. They do not find this signature there. Since the signature was created based on two patients, it is somewhat concerning that they do not see it in published literature. I assume that other scRNAseq datasets of BCC have been published. For sure there are datasets of other skin cancers, most prominently Melanoma. I would like to see a rigorous analysis of the TREM1 and BIT signatures in additional datasets.

16. In page 11 the authors say: “the 16- gene TREM1 myeloid signature score was markedly higher in BCC patients ... but even higher in BCCs with more abundant BIT tumor epithelium”. This is presented as new results, as if the authors analyzed more patients. However, if I understand correctly, the authors did not analyze additional BCC patients, and as such the results regarding the BIT high and low tumors are expected since this is how this signature was derived. I suggest that the authors tone down this statement or remove it altogether.

17. Fig. 7E: How were the thresholds for APO- and APO+ chosen? In the scatter it looks quite arbitrary.

18. In Supp. Figure 7A, less than 10% of the cells are EPCAM+. Since these are organoids, I would expect the majority of cells to be EPCAM+. Can the authors explain these results?

Minor notes:

1. The abstract uses the acronym BCC without explaining it.
2. Fig. 2g is not convincing. The authors should consider removing it.
3. Missing citation for anti-tumor properties of inflammation. (Introduction, second paragraph)
4. Missing citation for the following claim: " Sensitive, sonic hedgehog (Shh) pathway-driven BCC epithelium arises through the ability to recruit and instruct TREM2+VCAM1+ myeloid cells into a self propagating phenotype that allows mutual proliferation and tumor growth with a dearth of effector lymphocytes present in the local environment"
5. Missing citation for the following claim: "IL1 and OSM which are sufficient to induce BIT in murine BCC allografts and are reminiscent of autoinflammatory skin disorders like pustular psoriasis that are associated with neutrophilic infiltrates".

Reviewer #5:

Remarks to the Author:

Rebuttal to Reviewer Critiques

We thank all the reviewers for their strong and positive support for our work and for their insightful comments that should strengthen the manuscript and enhance its impact. We have endeavored to address all comments through additional experiments, analysis of additional datasets, and providing clear rationales. We have supplied data/figures within this point-by-point response and have indicated components that have been incorporated into the manuscript. All changes within the manuscript have been tracked for reviewers' convenience.

Reviewer #1, expertise in scRNA-seq, scATAC-seq, ST, skin cancer TME (Remarks to the Author):

In this manuscript, Li et al. profiled human skin basal cell carcinoma (BCC) samples (n = 4 total, consisting of n = 2 naïve and n = 2 smoothed inhibitor treated) with scRNA-seq, scATAC-seq, multiplexed imaging (CODEX, 41-protein panel), and spatial transcriptomics (10X Visium). Their analysis identified a novel subpopulation of tumor cells (or a tumor state) that they describe as basal to inflammatory transition (BIT), which is enriched in SMOi treated samples. BIT is marked by CHI3L1, TAGLN, ITGAV, and VCAM1 and seems to downregulate hedgehog pathway signaling (lower GLI expression). Integration of scATAC-seq data highlighted enrichment in NFKB family TF motifs within open chromatin of BIT cells. CODEX demonstrated that BIT cells are enriched at the superficial aspect of BCC tumors, but distinct from basal to squamous transition (BST) cells, a drug-resistant state that authors identified in earlier work. BIT cells are adjacent to tumor stroma consisting of myeloid cells such as neutrophils and macrophages. Conversely, BST cells are located more toward the tumor core and away from stroma. Authors then performed cell-cell communication analyses using scRNA-seq data and CellChat. This highlighted TREM1+ myeloid cell communication to BIT cells through IL1 and OSM. Visium data appeared consistent with CODEX findings, including BIT high cells being more superficially located than BST cells, with BIT cells in the vicinity of TREM1 myeloid cells. Finally, authors experimentally validated some of these findings using mouse BCC cell lines, which demonstrated that RNA-seq of combined IL1A+OSM treated BCC cells upregulated genes from the BIT signature, with ATAC-seq data highlighting enrichment of NFKB1 motifs within open chromatin after IL1A+OSM treatment. CUT&RUN with anti-NFKB1 supported direct binding of NFKB1 to the CHIL1 enhancer. Authors additionally performed in vivo experiments with intratumor injection of IL1A+OSM, which upregulated CHIL1. Human patient-derived organoids were able to resist SMOi-induced apoptosis in the presence of IL1+OSM. Overall, these studies identified a novel drug-resistant tumor state (BIT) distinct from the already established BST SMOi-resistant state, which seems at least partially regulated by OSM and IL1A that stem from TREM1+ myeloid cells. This study nicely combined several different single-cell and spatial profiling approaches of human samples to generate hypotheses regarding cell-cell communication in the TME, which were validated experimentally in both mouse and human models of BCC. The study could benefit from additional rigor of some analysis and experiments to more convincingly

support their claims. I detail some concerns below.

We very much thank Reviewer #1 for the positive and kind words regarding our manuscript and agree that our work furthers our understanding of cell-cell communication between BIT tumor state and TREM1+ myeloid cells within the tumor microenvironment.

Comment #1-1. I had trouble following how many samples were profiled for each technique. Based on Fig. 1a, 5 tumor samples were profiled for scRNA-seq and/or scATAC-seq, but at most 4 samples seem to be shown at a time. It is unclear if only some samples were profiled with both scRNA-seq and scATAC-seq or if the scATAC-seq were distinct samples. CODEX data is shown for 2 samples (1 in main figure, 1 in supplement), yet it seems 4 were profiled and analyzed. Same goes for the Visium data.

Comment #4-1. The paper is very vague on the number of samples that they had, which of these underwent which analytical assay and which of these is presented in each figure/panel. I would like to see a supplementary figure/table summarizing this information. It should also be displayed more clearly in figure 1.

Response #1-1/Response #4-1: We thank the reviewers for these comments. In the revised manuscript we provide a summary table of human BCC samples used for analysis in this manuscript and their corresponding profiling techniques in Supplementary Figure 1a.

a

Patient Tumors used for main figure analysis

Patient	(Haensel et al)					(Kuonen et al)						(Yao et al)					
	HuBCC1	HuBCC2	HuBCC3	HuBCC4	HuBCC5	HuBCC6	HuBCC7	HuBCC8	HuBCC9	HuBCC10	HuBCC11	HuBCC12	HuBCC13	HuBCC14	HuBCC15	HuBCC16	HuBCC17
scRNAseq	Yes	Yes	Yes	Yes	Yes	Yes	Yes	Yes	Yes	Yes	Yes	Yes	Yes	Yes	Yes	Yes	Yes
scATACseq	Yes	No	No	No	No	Yes	No	No	No	No	No	No	No	No	No	No	No
Visium	Yes	No	No	No	No	Yes	No	No	No	No	No	No	No	No	No	No	No
Tumor sample for CODEX	1	0	0	0	0	3	0	0	0	0	0	0	0	0	0	0	0

Comment #1-1a. How universal are these findings (BIT in consistent areas across all samples) and how do they vary across patients in spatial data? Please show data from all samples to provide clarity.

Comment #4-2. The major limitation of the human work is that for the most part it is based on 2-4 patients, depending on the assay. This is really a very low number to say anything about a subject as complex as the tumor microenvironment. While I understand that it is difficult to obtain many samples, and profile them with a large variety of technologies, perhaps the authors could suffice with a lower-plex IHC or IF panel to examine the existence and location of BIT in additional samples.

Response #1-1a/Response #4-2: As described in the manuscript, we find heterogeneity in the occurrence of BIT across patient samples with an incidence of about 1/3 of patient

tumors (Figure 1g). We performed in-depth profiling only for those tumor samples with relatively higher proportions of BIT populations in our initial scATAC-seq analysis, Visium analysis and CODEX while we left out the tumors with low proportions of BIT populations as those tumors lack an identifiable BIT cluster from scRNA-seq.

In the revised manuscript, we expand our analysis to include additional human BCC tumors from different datasets to demonstrate the heterogeneity in TS3/BIT abundance across individuals. We also provided summary statistics for spatial CODEX data across all 4 samples to support the claim of spatial proximity between neutrophils and macrophages to BIT neighborhood along with CODEX imaging data for an additional sample.

While our study is limited by sample size, the findings generated from multiple modalities coherently corroborate one another and points towards consistent insights into BIT as an indisputable tumor cell state transition that arises through extensive crosstalk with proximal TREM1 myeloid cells.

In addition, our spatial data (both CODEX and Visium) extensively profiles entire tumor tissue sections in huge areas as whole slide images rather than individual regions of interests (ROIs) which are limited in capture area and dimension. We think such advantage in spatial modality enables us to have a representative overview of the heterogeneity of tumor spatial neighborhoods, making our datasets uniquely well-positioned to provide insights into the global orchestration of complex tumor microenvironments (Fig 3e, supplemental figures 3c and d) that would otherwise be lost in ROI-based analysis.

Comment #1-1b. Nearest distance analyses and density plots also seem to be only done a subset of samples (Fig. 3h, S3d).

Response #1-1b: We thank the reviewer for the question and would like to clarify that the nearest distance analyses and density plots were done on all four CODEX samples – distances from all four samples for each tumor category were combined in a single plot.

We have also furnished Wilcoxon summary statistics across all four CODEX samples in Supplementary Figure 3. We have made such clarifications in our Figure 3 legends.

Comment #1-1c. Authors identified 17 cell types in CODEX (Fig. S3a), but most of the time, only a subset of them are shown in each CODEX sample (e.g., from Fig. 3d). Does each sample only contain a subset of identified cells? Are some cell types not present in certain samples?

Response #1-1c: We thank the reviewer for the question. All 17 cell types were present across all samples, the inter-tumor heterogeneity lies in the proportions/abundance of each cell type. It's hard to visualize all the 17 cell types on the tissue in one image, due to the constraint of color choices. For clear visualization purposes we could only display a maximum of 6 cell types in a single image to be discernible by human eyes. We have provided a cell type composition plot to illustrate the cell types identified for each sample in Supplementary Figure 3.

Comment #1-2. The BIT signature genes shown in Fig. 1f contain some genes that resemble epithelial to mesenchymal transition (EMT) or partial EMT, such as LAMB3, LAMA3, LAMC2, TGFBI among others. Yet, authors were quick to call this BIT based on implication of active NKFB signaling. Was GO analysis performed on the signature to support only an “inflammatory” state? Or are additional states present, i.e., some mixing of states happening? In Fig. 4b-c (CellChat analyses) TGFb (inducer of EMT) is prominently active, but authors seemed to ignore this.

Response #1-2: We thank the reviewer for this insightful question about the distinction between BIT and full EMT, one we also find intriguing. We include more detailed analysis and discussion in our revised manuscript to distinguish BIT from epithelial to mesenchymal transition (EMT). We have performed detailed analysis of epithelial cells and mesenchymal cells (fibroblasts) in naive human BCCs and identified a separate EMT

cluster that transits from epithelial state to mesenchymal state. The EMT cluster is distinguished from fibroblasts based on its high expression level of Keratin 14 and absence of fibroblast markers PDGFRA or PDGFRB. Meanwhile, the EMT cluster is high in expression of mesenchymal marker Vimentin and overall expression of EMT signature genes. While BIT tumor cluster also has slightly elevated expression of genes that are part of the EMT signature, its EMT signature score is way lower than the actual EMT cluster. Consistently, BIT tumor has higher expression of CDH1 (coding for E-cadherin) and lower expression of Vimentin compared to the EMT cluster. We have added this data into our revised manuscript Supplementary Figure 1.

We have also performed GO analysis on the TS3 signature as per reviewer's request and included the data in Supplementary Figure 1.

In addition, we also looked into our CODEX imaging data (Supplementary Figure 3a) to evaluate protein level expression of EMT markers in human BCCs. Consistent with our single cell data, we observed high expression level of E-cadherin and low expression level of Vimentin in BIT tumor, at levels comparable to other tumor states, suggesting BIT tumor

is largely epithelial in nature despite upregulation of some genes shared in EMT signature.

Further, across our spatial data we have showed that BIT tumors are spatially enriched in superficial aspect of BCC tumors, maintaining tight cell-cell contact and cell adhesion through integrins, inconsistent with an EMT phenotype where cells lose cell-cell contact and migrate/infiltrative into invasive fronts deep in the tumor. **We have added the discussion of this data in the revised manuscript.**

We have also evaluated BIT (TS3) signature in additional BCC datasets from BCC tumors that were either highly infiltrative in nature and at the invasive front (likely undergone EMT) (Kuonen et al) or highly inflammatory in nature (Gorlin Syndrome BCCs from Yao et al). We found that BIT (TS3) signature was highest amongst Gorlin Syndrome BCCs that are highly inflamed, yet not particularly elevated amongst infiltrative BCCs. Overall, scoring of BIT (TS3) signature across multiple BCC datasets provides stronger support for the implication of the signature in tumor inflammation rather than EMT. **We have added this data along with its discussion in the revised manuscript.**

We also thank the reviewer for pointing out the role of TGFb in BCC. Earlier work from our group (Kuonen et al, PMID: 29758283) has delineated in detail the role of TGFb in promoting invasion in basal cell carcinoma, a phenomenon not observed in BIT tumor state identified in this manuscript. We performed an additional experiment to investigate the role of TGFb in BIT induction in our mouse BCC cell line model. Consistent with our previous work, treatment of BCC cells with recombinant TGFb protein alone was insufficient to induce BIT → further supporting that BIT is an inflammatory tumor epithelial state driven by Il1 and Osm ligands rather than an EMT state driven by TGFb signaling.

Comment #1-3. Fig. 4h, can authors employ statistical testing on the BIT low vs. high tumors to show a significant difference in TREM1 myeloid cells?

Comment #4-11a. These are only 4 patients, yet the authors group them in 4h to the two groups and just show a single bar graph for each one. They should show a bar graph for each one of the patients.

Response #1-3/Response #4-11a: We thank the reviewers for the comments and agree that showing a bar graph for each individual patient would be a better way to present this data given the sample size. We have made the change as requested in Fig 4h.

Comment #1-4. In IL1A+OSM recombinant protein experiments, how much of the BIT signature is upregulated by IL1A+OSM? It looks partial at most, as some of the genes I mentioned above that were found in human data (LAMB3, TGFBI, etc.) are not shown in Fig. 6c.

Response #1-4: Owing to the differences between mouse and human BCC inflammation (i.e. Figs. 6 and 7), we have found that the murine system does not fully recapitulate the human BIT signature. We acknowledged this limitation in our discussion and are actively investigating the basis of this difference for future publications.

Comment #1-5. Fig. 6e, GSEA plots look very “off.” Authors should double check that GSEA was done correctly.

Response #1-5: The results of GSEA were presented using Barcode Enrichment Plots, which plot the positions of one or two gene sets in a ranked list of statistics. The statistics are ranked left to right from smallest to largest. The ranked statistics are represented by a

shaded bar or bed, and the positions of the specified subsets are marked by vertical bars, forming a pattern like a barcode.

Barcode plots are often used in conjunction with gene set tests, and show the enrichment of gene sets amongst high or low ranked genes. They were inspired by the set location plot of Subramanian et al (2005), with a number of enhancements, especially the ability to plot positive and negative sets simultaneously. Barcode plots first appeared in the literature in Lim et al (2009). More recent examples can be seen in Liu et al (2014), Sheikh et al (2015), Witkowski et al (2015) and Ng et al (2015).

Comment #1-6. There is a discrepancy between human and mouse data, as CHI3L1 appears to be the human BIT marker, while CHIL1 is shown for any mouse BCC experiments. Authors should address this difference.

Response #1-6: We thank the reviewer for the comment and would like to clarify that CHIL1 is the mouse equivalent of human CHI3L1 gene. They are equivalent genes coding for the same functional protein with different names in mouse versus human.

Comment #1-7. I do not see any IgG antibody control CUT&RUN data to demonstrate specificity of the NFKB1 antibody (Fig. 6h).

Response #1-7: We thank the reviewer for the comment and have added the sequencing track for IgG antibody control into Figure 6g to demonstrate specificity of the NFKB1 antibody. Additionally, Specificity of the NFKB1 antibody is strongly supported by motif enrichment analysis shown in Supplementary Figure 6f where there is statistically significant enrichment of NFKB family motifs among the binding sites of NFKB1 antibody.

Comment #1-8. Authors' experiments show that IL1A+OSM are at least partially sufficient to induce BIT and NFKB1 genome binding and activity inferred through motif analyses, but experiments do not show whether NFKB1 is required for BIT. I understand that the models are limited in that they do not spontaneously recapitulate BIT, but it would be more convincing to show a functional role of NFKB1 through KO or KD experiments.

Response #1-8: We thank the reviewer for the comment and have performed additional experiments using our mouse BCC cell line model to demonstrate BIT induction and reversibility by Il1a and Osm is dependent on NFKB and STAT3 through treatment with NFKB inhibitors JSH23 or WFA or STAT3 inhibitor Ruxolitinib. Our data shows that pretreatment of BCC cells with NFKB inhibitors JSH23 or WFA or STAT3 inhibitor Ruxolitinib was able to rescue BIT induction by Il1a and Osm, demonstrating functional involvements of NFKB1 and STAT3 in BIT induction.

Comment #1-9. While I note that raw sequencing data should be deposited in protected databases such as dbGAP, processed data such as counts matrices (or cell ranger outputs) for scRNA-seq/scATAC-seq should be included in the GEO submission, which is currently missing. Similarly processed spatial data should be deposited as well (at least for 10X Visium). It would be nice if authors made CODEX data available somehow as well.

Response #1-9: We thank the reviewer for the comment and will upload the processed sequencing data and processed spatial data with the revised manuscript.

Minor comments:

1. P. 13, variations should be variance if referring to PCA

We have changed the wording in our manuscript text.

2. Fig. 7f should be shown as paired samples and statistics applied.

We have statistics applied for Figure 7f and shown in Figure 7g.

Reviewer #2, expertise in CUT&RUN and chromatin biology (Remarks to the Author):

Summary: In this manuscript the authors have applied single-cell RNA-seq and ATAC-seq, the visium platform to perform spatial transcriptomics, and multiplexed imaging on human basal cell carcinoma samples, to define a new tumor microenvironment which the authors have termed BIT (basal to inflammatory transition). The authors use their single-cell assays to define a BIT signature and find that NFKB is involved in BIT. The authors suggest that BIT is initiated in a special immune-cell infiltrated inflammation site within tumors, and may contribute to drug resistance to the smoothed inhibitor.

Strong points: The multipronged approach taken by the authors to characterize heterogeneity in basal cell carcinoma is commendable. The authors have used elegant tools to analyze and identify a new microenvironment. This is potentially interesting and important in finding new therapeutic avenues. The comments below can strengthen the conclusions stated in the manuscript.

We thank the reviewer for the positive comments regarding our manuscript and are gratified to hear that they find our work important and appreciate the multi-modal techniques that we have undertaken to deeply profile the novel tumor microenvironment.

Major concerns:

Comment #2-1. The authors state throughout the manuscript that BIT emerges within a specialized inflammatory environment. However, how BIT form in tumors has not been thoroughly characterized in the current manuscript.

Response #2-1: We thank the reviewer for this comment although we respectfully disagree with the reviewer's notion that the current manuscript has not adequately characterize how BIT form in tumors.

First, using a combined single-cell and spatial approach, we have gone into detail characterizing this inflammatory transition in terms of tumor epithelial pseudo-time transcriptomic trajectory, using chromatin accessibility profiles to identify the NFKB transcription factor family as the main drivers responsible for the transition, and delineating the spatial localization of tumor cells with BIT signature in the tissue context.

We next identified the causal effect for the signature as being cytokines secreted from TREM1 myeloid cells that surround the tumor and cause activation of NFKB transcription in cancer cells. We subsequently endeavored to recapitulate this inflammatory signature by treating mouse and human BCC cells with Il1a and Osm cytokines and show that the newly acquired BIT phenotype leads to reduced apoptosis following SMO inhibitor treatment, the first line of therapy for advanced BCC patients.

In the revised version, we now include additional datasets and both in vitro and in vivo experiments with mouse and human tumors to elaborate the mechanistic basis of BIT, and how it differs from EMT. We believe our work has enabled us to provide a clear model

regarding the formation of BIT as delineated in our Figure 7h.

Comment #2-2. With the single-cell RNAseq data, the authors have defined a tumor state 3, which is claimed as being enriched by drug treatment. Firstly, this is only supported by Fig 1c. Secondly, the conclusion is based on the comparison of two treated tumor samples and two naïve tumor samples. It is understandable that applying all the single-cell approaches to a large number of samples is not feasible. The authors could change the title of Figure 1 (which currently states ‘Integrated single cell analysis of human BCC tumors identifies a novel tumor state (TS3) enriched by drug treatment’), since majority of the figure describes analysis from tumor samples that are not classified as treated or untreated.

We thank the reviewer for this suggestion. However, the motivation of the study comes from analysis of tumor populations enriched in drug treated tumors and their origin in pre-existence in naïve tumors.

Comment #2-3. The authors have used IL1 and OSM treatment in mouse BCC cells and performed RNA-seq, ATAC-seq, and CUT&RUN for NFKB1. The analyses of these datasets can be elaborated. For example, the authors can include heatmaps to show genome wide changes in chromatin accessibility and NFKB1 binding. The authors can also define clusters of genomic sites based on genes forming BIT v/s BST signature to strengthen their claim that BIT is distinct from BST.

Response #2-3: We thank the reviewer for this helpful suggestion and have performed more detailed analyses as requested. We have added the new data into Supplementary Figure 6.

Comment #2-4. Figure 7 convincingly shows that IL1 and OSM treatment reduces the sensitivity of SOMi treatment. However, the authors have not characterized how the expression of BIT markers changes upon SOMi treatment in IL1/OSM + and – organoids. The authors could perform RNAseq, ATAC-seq and NFKB1 CUT&RUN in the organoid system to dissect how IL1+OSM leads to resistance.

Response #2-4: We thank the reviewer for this comment, but our limited human patient sample availability precludes such extensive genomic profiling of patient-derived organoids. Our mouse organoid model allowed us to demonstrate the upregulation of BIT marker *Chil1* by qPCR but have provided additional in-depth profiling of the genome- and transcriptome-wide changes upon ligand treatment through RNA-seq, ATAC-seq and CUT&RUN in our mouse BCC cell line. In the revised version, we included the transcriptomic changes in our mouse organoids upon ligand treatment in Supplementary Figure 7, which demonstrate consistent genome-wide changes as observed in our mouse BCC cell line. This dataset will be useful for the cancer community to benchmark BIT-associated changes.

Minor concerns:

1. In line: Notably, among BCC tumor states, tumor state 3 was enriched in SMOi-treated BCC tumors relative to naïve BCC tumors (Fig. 1c, Supplementary figure 1d). Supp 1d does not support this statement.

We have edited the figure-callout text accordingly.

2. Some of the colours used to highlight the different cell-populations in the CODEX image (Fig 3) are very similar. Eg. BIT and epidermis in 3e. If possible the authors should use more contrasting colours.

We have updated the colors as requested.

3. In the discussion, the authors state 'and we now extend our initial observations through quantitative spatial expression analysis to confirm an optimal distance of 2-3 cell diameters (150-250 μm) from the microenvironment prior to the onset'. Where is the measurement coming from? It was not described in the results section.

We have removed the wording to avoid confusion.

Reviewer #3, expertise in basal cell carcinoma development, signalling and models
(Remarks to the Author):

In this article, Li et al. characterize a novel inflammatory signature enriched in therapy-resistant basal cell carcinoma (BCC). Since BCC is the most common human cancer, the research presented in this study is highly relevant and significant. In addition, due to the lack of effective therapeutic options for advanced BCC, finding new pathways to target tumor resistance is imperative. Despite BCC being one of the most mutated cancers, it is usually not highly immunogenic, and immunotherapy is only effective in a subset of patients. Characterization of the immunomodulatory events happening in BCC in human skin will help us better understand how these tumors develop and progress beyond the regulation of Hedgehog signaling.

The article focuses on a novel basal-to-inflammatory transition in cancer cells present in BCC and enriched in therapy-resistant tumors. The authors go into detail characterizing this inflammatory transition and the localization of tumor cells with this signature in the tissue context, comparing it to a previous resistance mechanism identified by the same group (called basal to squamous transition). They identify the causal effect for the signature as being cytokines secreted from myeloid cells that surround the tumor and cause activation of NF κ B transcription in cancer cells. Then, they recapitulate this inflammatory signature by treating mouse and human BCC cells with IL1 α and OSM cytokines and show that the newly acquired phenotype leads to reduced apoptosis following SMO inhibitor treatment, the first line of therapy for advanced BCC patients.

The study is novel since it defines a previously unknown inflammatory signature unrelated to other skin conditions like eczema and psoriasis that could be involved in therapeutic resistance. The detailed analysis of human patient samples also makes the study highly relevant. The article presents complex data analysis in a way that is easy to understand and follow, and all methods and figures are explained in detail.

We thank the reviewer for the strong and positive support for our manuscript and are gratified to hear that they find our work novel and highly relevant.

Some minor additional data and experiments might make the manuscript and the point stronger. Comments:

Comment #3-1. Although it is presented that the BIT signature can lead to therapy resistance by blocking apoptosis following SMO inhibitor treatment, no evidence is presented regarding the mechanisms by which this resistance might happen. Does cytokine treatment influence cell proliferation or only apoptosis resistance? Is there a change in proliferation or apoptosis in the tumors shown in Fig. 7a?

Response #3-1: We thank the reviewer for the comment. Our data from patient-derived organoids in Figures 7e-g demonstrated that upon ligand-induced BIT, there was reduced apoptosis following SMO inhibitor treatment based on APO-tracker green

staining (readout for apoptosis). We are actively investigating the exact mechanism underlying the reduced apoptosis for future publications.

Comment #3-2. One critical aspect is whether the BIT phenotype depends on the continuous activation of NFKB signaling. Is the phenotype reversible or stable? This key detail would determine if BIT can be therapeutically targeted. Could combining cytokine or NFKB inhibitors with SMO inhibitors revert the resistance phenotype caused by Il1a and OSM? Showing that the phenotype is reversible by drugs and/or cytokine withdrawal could be easily done in the mouse BCC cell line.

Response #3-2: We thank the reviewer for the suggestions and have conducted additional experiments to investigate whether BIT phenotype is reversible by drugs and/or cytokine withdrawal. Importantly, we demonstrated that the BIT phenotype was reversible upon cytokine withdrawal or antagonism. Consistently, pretreatment of BCC cells with NFKB inhibitors JSH23 or WFA or STAT3 inhibitor Ruxolitinib was able to rescue BIT induction by Il1a and Osm, supporting that BIT can be therapeutically targeted.

Comment #3-3. In line with the previous point, it is unclear if the increased number of BIT cells in therapy resistant BCC is due to an expansion of the BCC BIT cell population or an increased recruitment of TREM1 myeloid cells after treatment. So, does therapy expand previously existing BIT BCC cells, or does it lead to TREM1 myeloid cell recruitment that preserves and induces additional BIT BCC cells? It might be a difficult question to answer, but is it possible to analyze the proportion of TREM1-positive myeloid cells in naïve vs therapy-resistant BCC? This information is presented in suppl. Fig 5i and j for other cancer types but not for BCC.

Response #3-3: We thank the reviewer for the question and have performed additional analysis on myeloid cells in sensitive vs therapy-resistant BCCs. We did not observe a significant difference in TREM1 myeloid proportions among responders vs non-responders.

We have added this data in Supplementary Figure 5.

Based on our analysis (Figures 6 and 7), we conclude that BIT formation derives from host inflammatory response that allow enrichment during drug treatment. While therapies enrich for pre-existing BIT BCC cells prior to drug treatment, the local inflamed BIT microenvironment/niche has already enabled BIT cells to crosstalk with TREM1 myeloid cells in naïve conditions prior to any drug treatment. Instead of inducing further myeloid cell recruitment, drug treatments enrich for the BCC cells that have already undergone BIT transition in the specialized inflammatory milieu. We include additional discussion in the revised manuscript.

Comment #3-4. What would be a possible mechanism by which activation of NFKB leads to a loss of Hedgehog signaling? Fig. 5e shows a reduction in Hedgehog signature in cells, but it is not clear what precise components of the pathway are altered. Is there a reduction in the expression of Hedgehog components or a loss of transcriptional activity of Gli? This could be easily analyzed by looking at the expression of Hedgehog signaling components in the RNAseq data and the Gli binding sites in the ATAC-seq.

Response #3-4: We thank the reviewer for this helpful suggestion and have examined more closely components of the hedgehog pathway that were altered upon ligand treatment. We think the reduction in the expression of various hedgehog components collectively contributed to the suppression of Hedgehog signaling. Notably, we identified downregulation of both neuropilin 1 (Nrp1) and neuropilin 2 (Nrp2), which are known to play important roles as positive regulators of Hedgehog signal transduction (Hillman *et al.*, 2011, PMID: 22051878). In addition, Gai family of GPCR heterotrimeric G-proteins (encoded by *Gnai1*, *Gnai2* and *Gnaz* genes) were downregulated upon ligand treatment, which may have contributed to additional suppression of Hedgehog signaling through enhanced protein kinase A activity (Riobo *et al.*, PMID: 16885213, Hammerschmidt *et al.*, PMID: 8598293, Epstein *et al.*, PMID: 8787761). We have added the new data into Supplementary Figure 6.

Comment #3-5. The BST cells are located close to the skin surface. Could there be a correlation between the loss of epithelial barrier function and BIT acquisition? For example, authors could analyze if this signature is also present in wound healing datasets. Is it known whether vismodegib treatment leads to skin barrier dysfunction that could increase the recruitment of myeloid cells?

Response #3-5: We thank the reviewer for this insightful comment that we have actively investigated by analyzing scRNA-Seq datasets of skin diseases that have lost barrier function (Fig. 5g) and find no evidence of BIT. Our data in Figure 5g shows that BIT associated TREM1 myeloid signature was low among eczema and psoriasis patients despite the loss of epithelial barrier function associated with those skin conditions, whereas BIT associated TREM1 myeloid signature was higher in cancers that are more closely associated with inflammation (colorectal, ovarian and thyroid cancers), supporting the notion that the specialized inflammatory niche is tumor specific.

We have also investigated additional available wound healing datasets to determine if general wound healing promotes BIT. Among diabetic foot ulcer patients, we did not observe BIT induction in either healing or non-healing wounds – again supporting the notion that the specialized inflammatory niche is tumor specific.

In addition, we do not think the loss of epithelial barrier function was induced by vismodegib per se, as no reports of adverse drug effects of such kind have been made to FDA regarding the clinical use of vismodegib.

We are currently in the process of investigating under a different cover the correlation between Il1 and BIT formation and the host genetic basis for BIT predisposition.

Reviewer #4, expertise in multiplex imaging (Remarks to the Author):

Review for Li et al. Basal-to-inflammatory transition and tumor resistance via crosstalk with a pro-inflammatory stromal niche

Li et al. profile several treatment naïve patient samples using scRNAseq, scATAC, CODEX and Visium. They identify a tumor epithelial cell state, basal to inflammatory transition (BIT). They show that BIT is spatially localized in a distinct pro-inflammatory niche near TREM1 myeloid cells. They further demonstrate in both cell lines and organoids that IL1 and OSM secreted by TREM1 myeloid cells activate NF-kB within the BIT tumor epithelium, leading to transcriptional upregulation of BIT markers. Finally, they perform preliminary experiments in vivo to demonstrate that the BIT population confers increased resistance to SMO therapy.

Overall, the paper is interesting and coherent, proposing novel tumor-stroma interactions in basal cell carcinoma with potential therapeutic implications. The authors acquired diverse data on the human samples and did in vitro and in vivo work to provide mechanistic context for their results. I have some concerns, primarily regarding the analysis of the human data. I believe that the paper will be of interest to the readership of Nature Communications after the authors address the following the comments:

We thank the reviewer for their positive support of the work.

Comment #4-1. The paper is very vague on the number of samples that they had, which of these underwent which analytical assay and which of these is presented in each figure/panel. I would like to see a supplementary figure/table summarizing this information. It should also be displayed more clearly in figure 1.

Refer to **Response #1-1/Response #4-1.**

Comment #4-2. The major limitation of the human work is that for the most part it is based on 2-4 patients, depending on the assay. This is really a very low number to say anything about a subject as complex as the tumor microenvironment. While I understand that it is difficult to obtain many samples, and profile them with a large variety of technologies, perhaps the authors could suffice with a lower-plex IHC or IF panel to examine the existence and location of BIT in additional samples.

Refer to **Response #1-1a/Response #4-2.**

Comment #4-3. The authors should better acknowledge the limitation of the small sample size in their work and tone down many of their statements. Examples include (but are not limited to):

- a. "Our analysis corroborated previous work demonstrating that BST tumor epithelium occurs centrally and compactly within tumor nodules and does not interface directly with stromal elements"
- b. "BIT tumor epithelium arises within a pro-inflammatory myeloid dominant community, spatially and cellularly distinct from the stroma-deplete BST community".

Response #4-3: We thank the reviewer for this comment and plan to include, as is now typical, a section acknowledging the limitations of our study. However, addition of this section does not detract from our claims supported by our multi-omic and spatial data (Fig 3e, Supplementary Figures 3d, e, f, and Fig 5c, d, e, f).

Comment #4-4. Legends throughout the paper are very concise and not informative. Many elements in the figure are not explained. Please provide more information. Examples include (but are not limited to):

- a. Fig. 1H: What do the dashed lines represent? Are these confidence intervals? It's not stated in the plot or legend.
- b. Fig3h, what does the vertical line mean
- c. Fig6c, what does log(baseMean) mean?
- d. Fig4k, why is the y-axis range is so low? What is shown there?

Response #4-4: We thank the reviewer for the suggestion and will provide more details in our figure legends.

Comment #4-5. Fig. 1 The authors define TS3 as a state that is associated with poor prognosis, but they do not provide any information regarding what this state is, other than saying that it involves expression of CHI3L1, TAGLN, ITGAV and VCAM1. What is TS3? What are the other states that the authors identify in the tumors?

Response #4-5: TS3 is a novel tumor epithelial state marked by key markers shown in Figure 1f, which we later characterized throughout our manuscript as a tumor epithelial state that has undergone transition towards a more inflammatory phenotype through its active crosstalk with proximal myeloid cells. Among the other tumor epithelial states, TS1 and TS2 are basal and undifferentiated in nature, TS4 and BST are tumor epithelial cells that are more differentiated and undergo squamous transition, while cycling and proliferative tumor states represent tumor cells undergoing cell cycling or in active proliferation respectively. We have added the above information into the text of the revised manuscript.

Comment #4-6. The authors saw a rise in NF-kB signaling in the 2 treated patients, suggesting a correlation with the rise of TS3. Since this finding is based on only two patients, can the authors check the other datasets with TS3 to see if they also have more NF-kB signaling?

Response #4-6: We thank the reviewer for the suggestion and have analyzed more single cell BCC datasets. Consistent with our in-house datasets, we indeed observed high TS3 signature along with high NF-kB signaling among the more inflammatory BCC tumors from Yao et al. We have included the new data in Supplementary Figure 2g.

g
Comment #4-7. Fig. 2A,B: The authors perform monocle pseudotime analysis, but is there any rationale to doing this? Pseudotime applies when examining a developmental process. Do the authors think that the compendium of states observed in the tumor are all a process of development? I would naïvely assume that some of the differences in states have to do with the different microenvironments of the tumor. What is the meaning of a starting state and end states in the context of a tumor? Why are these not just cooccurring states?

Response #4-7: We thank the reviewer for the questions. We indeed think that the compendium of states observed in the tumor epithelium are all a process of development (with the exception of cycling or proliferating tumor epithelial cells). Our pseudotime trajectory analysis was done exclusively on the tumor epithelial populations, excluding other stromal or immune cell types that are present in the tumor microenvironment. Hence the developmental trajectory shown indicates a tumor cell-intrinsic developmental trajectory that takes place exclusively within tumor epithelium. While the BIT developmental trajectory is heavily dependent on the inflammatory microenvironment proximal to the BIT tumor epithelium (which we elaborately dissected in Figures 3, 4, and 5), the BST developmental trajectory is largely stromal independent and mainly driven by tumor cell intrinsic AP-1 family of TF drivers (Kuonen et al 2021, Haensel et al. 2022).

The starting state (basal undifferentiated state) is the predominant tumor state in naïve nodular BCCs. The end states represent tumor evolution and pathway switching from Hedgehog signaling dependency to alternative signaling pathways. Such tumor cell state plasticity takes place either spontaneously or under drug selection pressure. The transitioned tumor states render BCCs resistant to targeted therapy vismodegib that targets Hedgehog pathway exclusively.

While basal tumor state is the predominant tumor state across all nodular BCCs, the presence and abundance of transitioned BST and BIT tumor states vary across BCC tumors, suggesting that BST and BIT tumor cell states do not universally co-occur across all BCCs but only occur in a subset of BCCs as a result of tumor cell state transition, which we deem as undeniably a developmental process.

Comment #4-8. Fig. 2D: It is unclear from the methods how the scATAC and scRNAseq data were merged. The UMAP in 2D, showing the merged data looks strikingly similar to the UMAP in 2C which was constructed only based on the scATAC data. How come?

Response #4-8: We have provided more details in our methods section regarding data integration. We would like to clarify that integration of scATAC-Seq data with scRNA-Seq data does not change the UMAP embeddings of the scATAC-Seq data but adds on a refined layer of RNA expression information to allow for refined clustering of cells.

Comment #4-9. Throughout figure 2 the authors do not provide meaningful information regarding the cell states that were derived from the scATAC or scATAC+scRNAseq. They simply refer to them as C1-C5 or “Tumor state1 – Tumor state5”. This is not very informative. What do I as a reader learn from this analysis? The authors should do the minimal level of analysis, which is to annotate the clusters that they find and relate it back to the literature.

Response #4-9: We thank the reviewer for the suggestion. Heatmaps of cluster markers of all cell types and tumor epithelial states C1-C5 have been included in Supplementary Figures 2d-e.

Comment #4-10a. Why is the UMAP in 4A different the one in figure 1E? The legend suggests that only a subset of the tumor cells were taken for the cellChat analysis. Why not all of them?

Response #4-10a: The tumor cells of UMAP in 4a are exactly the same as the ones in

Figure 1e and all tumor epithelial populations were included in CellChat analysis. The “subset” refers to tumor epithelial populations and myeloid populations included in panel 4a rather than all cell types. The goal of Figure 1e was to provide a refined examination of the tumor cell states at a high resolution, whereas the same tumor cells were more broadly clustered in Figure 4a into several major cell state categories. The difference in UMAP embeddings is due to the addition of myeloid cells in Figure 4a. With the addition of myeloid cell types, we lowered the resolution for tumor population clustering such that the “Tumor basal” cluster in 4a is a combination of “Tumor state 1” and “Tumor state 2” in Figure 1e which are basal and undifferentiated in nature; “BST” cluster in 4a is a combination of “BST” and “Tumor state 4” in Figure 1e which are more differentiated in nature (please refer to tumor state cluster markers in Figure 1f); “BIT” cluster in 4a matches “Tumor state 3” in 1e and “Proliferative” cluster in 4a is a combination of “Cycling” and “Proliferative” tumor states in 1e. Grouping tumor states into the major categories shown in Figure 4a provides the same insights in terms of cell-cell communication through CellChat analysis and allows for greater interpretability of results. We will reword the legend to avoid the confusion.

Comment #4-10b. Similarly, why are the values in the graphs in 4B and Supp. 4C different? For example, BIT in 4b is $\pm(1.5,2)$ whereas BIT in Supp. 4c is $\pm(2.1,2.2)$.

Response #4-10b: Figure 4b depicts CellChat analysis performed on tumor cell populations and myeloid cell populations only, while Supplementary 4c depicts CellChat analysis performed on all cell types at the global level. Hence the incoming and outgoing signaling strength plot would have different scales. The take-home message is that regardless of examining all cell types at the global scale or a more focused angle, BIT has the highest incoming and outgoing signaling strength.

Comment #4-11. In figure 4(h,i) the authors claim that the presence of the BIT state in the tumor cells correlates with TREM1 Myeloid cells. To show that, they classify 4 tumors into two high BIT (4.6% and 5.8%) and two low BIT (1.6% and 1.8%). They show that in the BIT high group the TREM1+ are $\pm 60\%$ of the myeloid cells, whereas in the BIT low they are $\pm 25\%$. I find this analysis problematic for several reasons:

Comment #4-11a. These are only 4 patients, yet the authors group them in 4h to the two groups and just show a single bar graph for each one. They should show a bar graph for each one of the patients.

Response #4-11a: We thank the reviewer for the suggestion and have made the change in Figure 4h accordingly.

Comment #4-11b. Similarly, for the UMAP of the Myeloid cells, the authors should show it colored by patients.

Response #4-11b: We thank the reviewer for the suggestion and have added the data in Supplementary Figure 4.

Comment #4-11c. Are any of these 4 patients the same as the two patients that were described in figures 1-3 4a-f? It is not clear enough from the text. If it is not the same patients, then there is a mistake in Figure 1A, and it should say that the paper had 8 patients (2 naïve, 2 drug resistant, 4 naïve for validation). If it is the same patients, then the analysis in 4k is misleading. If these are the same patients, then by definition we would expect to see e.g. IL1A in TREM1 Myeloids, because this is how the CellChat analysis found it. If this is an independent validation then it should be conducted on different patients.

Response #4-11c: Yes, the two patients that were described in earlier figures were among the 4 patients in this analysis. In Figure 4k, we indeed see elevated IL1A, IL6 and OSM in TREM1 Myeloids. Figure 4k provides important information on key marker genes for the different myeloid populations, corroborates important findings from CellChat analysis, and provides rationale for the validation experiments carried out in Figure 6a. We have provided a table of all patient samples and analysis done to clarify this to avoid the confusion.

Comment #4-11d. The changes in TREM1 myeloid cells are big (25%-60%), yet the changes in the BIT fraction are minuscule (2%-5%). How does this fit?

Response #4-11d: Myeloid cells represent a smaller proportion in absolute numbers than tumor cells in tumor tissue, so proportion changes in myeloid cell populations naturally fluctuate to a greater extent than proportion differences in tumor cells, especially in the context of local recruitment in the tumor microenvironment. We would like to respectfully disagree with the reviewer's notion that changes in the BIT fraction are "minuscule". We think the difference in BIT proportion between 2% and 5% of total tumor cells means a sea of difference in terms of myeloid recruitment, tumor microenvironment changes and subsequent treatment responses. We would like to point the reviewer to our spatial data in Figures 3, 5 and Supplementary Figures 3 and 5 to appreciate the spatial influence of BIT tumor which we believe goes beyond its mere abundance.

Comment #4-12. The authors suggest IL1 and OSM secretion from TREM1 Myeloid cells as important signals influencing BIT. They do not relate this at all to the literature. IL1 is known to induce inflammation in cancer in general and in skin cancer in particular. It was already suggested as a signaling pathway that can contribute to resistance mechanisms (e.g. DOI 10.1016/j.cyto.2022.155850 and 10.18632/oncotarget.12590). The same holds for IL6, which OSM is part of the IL6 family (e.g. DOIs 10.1158/0008-5472.CAN-14-2061). There is also work showing that blocking IL1 and IL6 increase anti-tumor activity and that NFkB is involved in this signaling pathway. The authors should put their results in the context of the existing literature.

Response #4-12: We thank the reviewer for the suggestion and have discussed our findings in the context of the literature mentioned above in our discussion.

Comment #4-13. Fig. 5A, same comment as above. Since these are only 4 patients, there is no reason to group them. The authors should kindly show the expression in each patient.

Response #4-13: We thank the reviewer for the suggestion and have made the change in Figure 5a accordingly.

Comment #4-14. Fig. 5: According to the authors, this analysis is “validating the spatial association between BIT tumor epithelium and the specialized TREM1 myeloid cell-associated inflammatory environment” (page 10). However, this Visium analysis is performed on the same patient on which the single-cell and CODEX analyses were performed. As such, it doesn’t provide a general validation for the association of these cell types across patients. The authors should tone down their statement to reflect the fact that this is an additional way to look at the same tumor.

Response #4-14: We appreciate the reviewer’s comment and have changed the wording “validating” to “reinforcing” to avoid confusion.

Comment #4-15. Fig. 5G: The authors compare their TREM1 signature to two datasets of healthy skin, eczema, psoriatic and alopecia areata. They do not find this signature there. Since the signature was created based on two patients, it is somewhat concerning that they do not see it in published literature. I assume that other scRNAseq datasets of BCC have been published. For sure there are datasets of other skin cancers, most prominently Melanoma. I would like to see a rigorous analysis of the TREM1 and BIT signatures in additional datasets.

Response #4-15: We thank the reviewer for this suggestion. We have gone ahead and expanded our analysis to include additional datasets where we compared myeloid cells from our BCCs to normal skin, inflammatory skin diseases and datasets from multiple cancer types from a pan-cancer study. We have also included a few other well-known datasets including alopecia areata and colon cancer. Our data in Figure 5g shows that BIT associated TREM1 myeloid signature was low among eczema and psoriasis patients despite the loss of epithelial barrier function associated with those skin conditions, whereas BIT associated TREM1 myeloid signature was higher in cancers that are more closely associated with inflammation (colorectal, ovarian and thyroid cancers). This new data supports the association of TREM1 signature with inflammation in a tumor-specific context rather than general inflammation or skin barrier dysfunction outside of tumor context.

We have also investigated additional available wound healing datasets to determine if general wound healing promotes BIT. Among diabetic foot ulcer patients, we did not observe BIT induction in either healing or non-healing wounds – again supporting the notion that the specialized inflammatory niche is tumor specific. We have added this new data to Figure 5 and Supplementary Figure 5.

Comment #4-16. In page 11 the authors say: “the 16-gene TREM1 myeloid signature score was markedly higher in BCC patients ... but even higher in BCCs with more abundant BIT tumor epithelium”. This is presented as new results, as if the authors analyzed more patients. However, if I understand correctly, the authors did not analyze additional BCC patients, and as such the results regarding the BIT high and low tumors are expected since this is how this signature was derived. I suggest that the authors tone down this statement or remove it altogether.

Response #4-16: Here we reported the results of benchmarking our TREM1 myeloid signature against the myeloid populations found in other inflammatory skin diseases. We think such benchmarking is crucial in distinguishing BIT associated inflammatory myeloid phenotype from inflammatory myeloid phenotypes observed in general skin inflammation outside of the tumor context, highlighting the tumor-specific aspect of this specialized TREM1 myeloid signature. We think the addition of datasets from multiple cancer types would be able to further strengthen this statement.

Comment #4-17. Fig. 7E: How were the thresholds for APO- and APO+ chosen? In the scatter it looks quite arbitrary.

Response #4-17: The thresholds were chosen based on relative staining intensities to stratify cells into APO high and APO low. The same threshold gate was applied across treatment arms to enable objective comparison between control condition and treatment condition.

Comment #4-18. In Supp. Figure 7A, less than 10% of the cells are EPCAM+. Since these are organoids, I would expect the majority of cells to be EPCAM+. Can the authors explain these results?

Response #4-18: All of these patient derived organoids have undergone vismodegib treatment which we believe greatly eliminate the majority of hedgehog-driven EPCAM+ tumor cells that are sensitive to the vismodegib treatment. Therefore the remaining 10% EPCAM+ cells indicate the resistant tumor cell fractions that have not responded to

vismodegib treatment.

Minor notes:

1. The abstract uses the acronym BCC without explaining it.

We have made the change in the revised manuscript.

2. Fig. 2g is not convincing. The authors should consider removing it.

We acknowledge the fact that genes coding for transcription factors typically suffer from drop-out in single cell studies and hence appear less striking in feature plots. Still, we think Figure 2g provides important information regarding differences in expression levels of the transcription factors of interest at the cluster level and decide to keep it in our figure.

3. Missing citation for anti-tumor properties of inflammation. (Introduction, second paragraph)

We have added the citation in the revised manuscript.

4. Missing citation for the following claim: " Sensitive, sonic hedgehog (Shh) pathway-driven BCC epithelium arises through the ability to recruit and instruct TREM2+VCAM1+ myeloid cells into a self propagating phenotype that allows mutual proliferation and tumor growth with a dearth of effector lymphocytes present in the local environment"

We have added the citation in the revised manuscript.

5. Missing citation for the following claim: "IL1 and OSM which are sufficient to induce BIT in murine BCC allografts and are reminiscent of autoinflammatory skin disorders like pustular psoriasis that are associated with neutrophilic infiltrates".

We have added the citation in the revised manuscript.

Reviewer #5 (Remarks to the Author)

Reviewers' Comments:

Reviewer #1:

Remarks to the Author:

I appreciate the authors' additional experiments, analyses, and clarifications. They have addressed all my concerns. Congratulations on this very nice study.

Reviewer #2:

Remarks to the Author:

The reviewers have addressed the concerns raised in the first round. I do not have additional concerns.

Reviewer #3:

Remarks to the Author:

Authors have appropriately answered my concerns and modified the manuscript according to the reviewer's comments. The new data and figures strengthened the manuscript.

Reviewer #4:

Remarks to the Author:

The authors have addressed most of my comments satisfactorily. A few small issues remain, which I think the authors should correct prior to publication.

Response #4-11c: The authors acknowledge that two of the four patients in figure 4(h,i) were also described in figures 1-3. As such, this is not an independent validation. The authors have added a Supp. Table to detail which patients underwent which analysis, but in this case, it is not enough since it is material for the results. They should additionally explicitly state in the text and annotate in the figure which patients were previously examined, and which present a new independent validation.

Response #4-17: The response is not satisfactory. Obviously, thresholds for APO- and APO+ were chosen to stratify cells into APO high and APO low. They still seem arbitrary, as there is no clear bimodal distribution that justifies them. The authors have not convinced me in their response that these were not specifically tailored to yield specific results. The authors should produce a supplementary plot showing where the thresholds fall on an expression histogram of APO.

Reviewer #5:

Remarks to the Author:

I co-reviewed this manuscript with one of the reviewers who provided the listed reports. This is part of the Nature Communications initiative to facilitate training in peer review and to provide appropriate recognition for Early Career Researchers who co-review manuscripts

REVIEWERS' COMMENTS

We thank all the reviewers for their strong and positive support for our work.

Reviewer #1 (Remarks to the Author):

I appreciate the authors' additional experiments, analyses, and clarifications. They have addressed all my concerns. Congratulations on this very nice study.

We thank Reviewer #1 for the positive and kind words regarding our manuscript.

Reviewer #2 (Remarks to the Author):

The reviewers have addressed the concerns raised in the first round. I do not have additional concerns.

We thank the reviewer for the positive comments regarding our manuscript.

Reviewer #3 (Remarks to the Author):

Authors have appropriately answered my concerns and modified the manuscript according to the reviewer's comments. The new data and figures strengthened the manuscript.

We thank the reviewer for the strong and positive support for our manuscript.

Reviewer #4 (Remarks to the Author):

The authors have addressed most of my comments satisfactorily. A few small issues remain, which I think the authors should correct prior to publication.

Response #4-11c: The authors acknowledge that two of the four patients in figure 4(h,i) were also described in figures 1-3. As such, this is not an independent validation. The authors have added a Supp. Table to detail which patients underwent which analysis, but in this case, it is not enough since it is material for the results. They should additionally explicitly state in the text and annotate in the figure which patients were previously examined, and which present a new independent validation.

We thank the reviewer for their positive support of the work.

We respectfully disagree with the reviewer's comment here as reviewer seemed to misunderstand the purpose of figures 4(h,i) here. Figure panels 4(h,i) present detailed analysis of myeloid populations of 4 BCC patients, in order to shed light on the specialized properties of the subset of myeloid populations that are associated with BIT. We have not claimed in our manuscript, nor do we intend to claim, that the purpose of figure panels 4(h,i) intend to serve as an independent validation dataset. Compared to

Figures 1-3 which have been an extensive examination of the BIT tumor epithelium, Figure panels 4(h,i) are key panels in the entire manuscript that shift the focus of discussion from BIT tumor epithelium itself towards BIT's associated myeloid populations in patients' tumor microenvironment. Therefore, none of the myeloid populations examined in Figure panels 4(h,i) were previously examined in Figures 1-3. A detailed summary table on the 4 naïve BCC patients examined in figure panels 4(h,i) is provided in Supplementary Figure S4f.

Response #4-17: The response is not satisfactory. Obviously, thresholds for APO- and APO+ were chosen to stratify cells into APO high and APO low. They still seem arbitrary, as there is no clear bimodal distribution that justifies them. The authors have not convinced me in their response that these were not specifically tailored to yield specific results. The authors should produce a supplementary plot showing where the thresholds fall on an expression histogram of APO.

We thank the reviewer for the comment and have provided the required plots in Supplementary Figure S7c. The expression histograms support the presence of a bimodal distribution in the expression levels of APO that guided the choice of thresholds.

Reviewer #5 (Remarks to the Author):

I co-reviewed this manuscript with one of the reviewers who provided the listed reports. This is part of the Nature Communications initiative to facilitate training in peer review and to provide appropriate recognition for Early Career Researchers who co-review manuscripts